# Placental *Streptococcus agalactiae* DNA is associated with neonatal unit admission and foetal pro-inflammatory cytokines in term infants

**Francesca Gaccioli** [1,2,11], **Katie Stephens** [1,11], **Ulla Sovio** [1,2,11], **Flora Jessop**[3], **Hilary S. Wong**[4], **Susanne Lager**[5], **Emma Cook** [1], **Marcus C. de Goffau**[6,7], **Kirsty Le Doare**[8], **Sharon J. Peacock** [9], **Julian Parkhill** [10], **D. Stephen Charnock-Jones** [1,2,12] ✉ & **Gordon C. S. Smith** [1,2,12] ✉

*Streptococcus agalactiae* (Group B *Streptococcus*; GBS) is a common cause of sepsis in neonates. Previous work detected GBS DNA in the placenta in ~5% of women before the onset of labour, but the clinical significance of this finding is unknown. Here we re-analysed this dataset as a case control study of neonatal unit (NNU) admission. Of 436 infants born at term (≥37 weeks of gestation), 7/30 with placental GBS and 34/406 without placental GBS were admitted to the NNU (odds ratio (OR) 3.3, 95% confidence interval (CI) 1.3–7.8). We then performed a validation study using non-overlapping subjects from the same cohort. This included a further 239 cases of term NNU admission and 686 term controls: 16/36 with placental GBS and 223/889 without GBS were admitted to the NNU (OR 2.4, 95% CI 1.2–4.6). Of the 36 infants with placental GBS, 10 were admitted to the NNU with evidence of probable but culture-negative sepsis (OR 4.8, 95% CI 2.2–10.3), 2 were admitted with proven GBS sepsis (OR 66.6, 95% CI 7.3–963.7), 6 were admitted and had chorioamnionitis (inflammation of the foetal membranes) (OR 5.3, 95% CI 2.0–13.4), and 5 were admitted and had funisitis (inflammation of the umbilical cord) (OR 6.7, 95% CI 12.5–17.7). Foetal cytokine storm (two or more pro-inflammatory cytokines >10 times median control levels in umbilical cord blood) was present in 36% of infants with placental GBS DNA and 4% of cases where the placenta was negative (OR 14.2, 95% CI 3.6–60.8). Overall, ~1 in 200 term births had GBS detected in the placenta, which was associated with infant NNU admission and morbidity.

*Streptococcus agalactiae* (Group B *Streptococcus*; GBS) is the most common cause of sepsis in the first week of life[1], and this is referred to as early-onset disease (EOD). GBS is present in the genital tract in about 20% of women, and, in the absence of intervention, about 1% of infants born to a colonized mother develop EOD[2,3]. An analysis of the global burden of disease in 2020 estimated that about 20 million pregnant women were colonized with GBS, 230,000 infants exhibited EOD and GBS accounted for around 50,000 stillbirths and 50,000–100,000

**Table 1 | Association between the presence of GBS DNA in the placenta and the risk of neonatal unit admission, classified by clinical assessment or histopathology**

| | Placental GBS | | Univariable analysis | | | Multivariable analysis | | |
|---|---|---|---|---|---|---|---|---|
| | Positive n (%) | Negative n (%) | OR | 95% CI | P | OR | 95% CI | P |
| **Neonatal unit admission** | | | | | | | | |
| Not admitted (N=686) | 20 (2.9) | 666 (97.1) | | | | | | |
| All admissions (N=239) | 16 (6.7) | 223 (93.3) | 2.4 | 1.2–4.6 | 0.009 | 2.3 | 1.4–4.5 | 0.02 |
| **By diagnosis** | | | | | | | | |
| No sepsis (N=60) | 2 (3.3) | 58 (96.7) | 1.1 | 0.3–4.6 | 0.85 | 1.0 | 0.2–4.4 | 0.99 |
| Possible sepsis (N=90) | 2 (2.2) | 88 (97.8) | 0.8 | 0.2–2.9 | 0.71 | 0.7 | 0.2–3.2 | 0.67 |
| Probable sepsis (N=80) | 10 (12.5) | 70 (87.5) | 4.8 | 2.2–10.3 | <0.0001 | 4.5 | 2.0–10.1 | 0.0003 |
| Proven GBS sepsis (N=3) | 2 (66.7) | 1 (33.3) | 66.6 | 7.3–963.7 | 0.003 | | | |
| **By histopathology** | | | | | | | | |
| No inflammation (N=186) | 9 (4.8) | 177 (95.2) | 1.7 | 0.7–3.6 | 0.19 | 1.5 | 0.6–3.3 | 0.36 |
| Chorioamnionitis (N=44) | 6 (13.6) | 38 (86.4) | 5.3 | 2.0–13.4 | 0.0002 | 5.3 | 2.0–14.1 | 0.0008 |
| Funisitis (N=30) | 5 (16.7) | 25 (83.3) | 6.7 | 2.5–17.7 | 0.0001 | 5.9 | 2.0–17.3 | 0.001 |

For univariable analysis, unadjusted odds ratio (OR) with Baptista–Pike mid-P 95% CI and chi-squared test P values are presented. Due to small numbers, Fisher's exact test two-sided P value is given for proven GBS sepsis. For multivariable analysis, OR, 95% CI and P values were estimated using logistic regression analysis, adjusted for maternal characteristics (age, BMI, smoking and marital status). Due to small numbers, multivariable analysis is omitted for proven GBS sepsis. Chorioamnionitis and funisitis were both present in 4 cases where the placenta was GBS DNA positive and in 19 of the cases when it was negative. The diagnosis for six cases of neonatal unit admission could not be confirmed due to missing information.

infant deaths[4]. However, EOD remains uncommon, affecting less than 1 in 1,000 births in high income countries. The incidence is higher in low- and middle-income settings, but reliable estimates are problematic due to incomplete ascertainment. We previously used metagenomic methods to determine whether bacterial DNA was present in the placenta at term and found that the only verifiable bacterial signal before membrane rupture and the onset of labour at term was GBS[5]. In this Article, we employed data and biological samples from a prospective cohort of >4,000 nulliparous women to determine whether the presence of GBS DNA in the placenta was associated with the risk of neonatal morbidity.

## Results

### Placental GBS and the risk of neonatal morbidity

We analysed data from our previously reported dataset in which GBS had been quantified in the placenta from 537 pregnancies[5]. In this discovery dataset the presence of GBS DNA was defined using either deep sequencing or PCR of the bacterial 16S ribosomal RNA gene followed by amplicon sequencing, as previously described[5]. In the new analysis of this study, we defined cases as admission of the infant to the neonatal unit (NNU) within 48 h of birth and for 48 h or more, as previously employed[6]. Given the high rates of NNU admission of preterm infants, we confined the analysis to 436 births at term (41 cases and 395 controls) (Extended Data Fig. 1 and Supplementary Table 1). Controls were defined as any term birth that did not fulfil the case criteria. Of the 436 infants born at term, 7/30 (23.3%) with placental GBS DNA and 34/406 (8.4%) without placental GBS DNA were cases (odds ratio (OR) 3.3, 95% confidence interval (CI) 1.3–7.8, P = 0.007). The association was similar following adjustment for maternal characteristics (adjusted OR 3.1, 95% CI 1.2–8.0, P = 0.02).

To validate this finding we performed a case control study using non-overlapping subjects drawn from the same cohort. Cases were defined as pregnancies where the infant was admitted to the NNU without time limits for admission or duration of stay. Controls were infants who were not admitted to the NNU. We excluded preterm births and cases and controls that had already been included in the discovery study described above, leaving 239 cases and 686 controls (Extended Data Fig. 1 and Supplementary Table 2). A power calculation indicated that we had >99% power to replicate the original finding with this sample size. We determined placental GBS DNA status

using an ultrasensitive polymerase chain reaction (PCR)–quantitative PCR (qPCR) assay targeted at a GBS-specific region of the 16S rRNA gene (Extended Data Figs. 2 and 3). Consistent with the results of the discovery study, 16/36 infants with placental GBS and 223/889 without GBS were admitted to the NNU (OR 2.4, 95% CI 1.2–4.6), and the association was similar following adjustment for maternal characteristics (Table 1).

### Septic and non-septic phenotypes of neonatal morbidity

We hypothesized that the risk of neonatal unit admission was causally explained by the presence of GBS in the placenta. This hypothesis predicts that the association would be progressively stronger as the evidence for neonatal sepsis and inflammation was stronger. Hence, we next performed secondary analyses of the validation case control study in relation to associated evidence for sepsis. First, we classified cases of NNU admission by clinical evidence, with classification performed blind to placental GBS DNA status. The infant's clinical record was reviewed and admissions were classified on an ordinal scale: no evidence of sepsis, possible sepsis, probable but culture-negative sepsis or proven GBS sepsis, with proven sepsis being defined on the basis of a positive postnatal culture result from a septic screen (Supplementary Table 3). Analysis of the 16 NNU admissions with placental GBS demonstrated that there was no association between placental GBS and the risk of NNU admission with no evidence of sepsis or only possible evidence (Table 1). However, there were strong associations with NNU admission with probable but culture-negative sepsis (OR 4.8, 95% CI 2.2–10.3) and proven GBS sepsis (OR 66.6, 95% CI 7.3–963.7).

We next classified admissions on the basis of histopathological examination of the foetal membranes and a cross section of the umbilical cord (Fig. 1), again assessed blind to placental GBS DNA status. Inflammation was defined on the basis of a previously described ordinal scale (stage 0, 1, 2 or 3) for chorioamnionitis and funisitis, respectively, and in both cases samples with stage 2 or 3 findings were defined as positive[7]. There was no association between placental GBS and the risk of NNU admission without evidence of chorioamnionitis or funisitis (Table 1). However, there were strong associations between GBS DNA in the placenta and NNU admission with chorioamnionitis (OR 5.3, 95% CI 2.0–13.4) and funisitis (OR 6.7, 95% CI 12.5–17.7). All associations were similar following adjustment for maternal characteristics (Table 1).

### Rates of placental GBS by genital tract colonization

Of the 925 participants in the second case control study, 467 (50%) had a report of a high vaginal swab (HVS) during the pregnancy. GBS DNA was detected in the placenta in 7/57 (12%) of participants with a positive HVS culture for GBS and 8/410 (2%) of those with a negative HVS culture for GBS. Of those who had no HVS culture performed during the pregnancy, 21/458 (5%) had placental GBS DNA detected. There was an excess of GBS-positive placentas in the first group when compared with the second and third ($P < 0.0001$ and 0.02, respectively; Fig. 2a).

### Validating the GBS DNA signal

We tested the possibility that GBS DNA was detected in the placenta due to contamination. The design and results of the no-template controls from the discovery case control study are reported in our previous paper[5]. In relation to the validation case control study, there was no temporal relationship between the presence of GBS DNA in the placenta and the week of delivery and there were no clusters of positive cases (Extended Data Fig. 4 and Supplementary Table 4). We compared the proportion of infants delivered by caesarean section, where the placenta is delivered abdominally precluding contamination during vaginal birth. A greater proportion of participants with a GBS DNA-positive placenta were delivered by caesarean section compared with those who were GBS DNA negative (47% versus 28%, respectively, $P = 0.01$) (Supplementary Table 5). The proportion of caesarean delivery was even higher (63%) in the GBS DNA-positive cases where the infant experienced neonatal morbidity related to sepsis (Table 2). This pattern is the opposite of what would have been predicted if GBS DNA was present in the placenta due to contamination at the time of vaginal birth. We also compared the strength of association between having a positive high vaginal swab in the antenatal period and the risk of the outcomes of interest. There were either no statistically significant associations or the associations were weaker than observed with the presence of GBS DNA in the placenta (Supplementary Table 6). Hence confounding by contamination of the placenta with GBS during vaginal birth is not a plausible explanation for our findings. This interpretation is consistent with the fact that the biopsy was taken from the interior of the placenta and was washed to remove surface contamination (Methods).

We next considered the potential for contamination during sample analysis. There was also no evidence of batch effects in relation to the conduct of the PCR–qPCR assay for placental GBS (Extended Data Fig. 5 and Supplementary Table 7). Moreover, all of the qPCR plates contained multiple negative controls, which were included at every step of the experimental workflow, and none of these (0/129 wells) were positive for GBS. To test the validity of the GBS DNA signal further, we analysed additional samples from the 36 placentas that were positive for GBS DNA and from 36 GBS DNA-negative placentas using a novel reverse transcriptase (RT)–qPCR assay for GBS 16S ribosomal RNA (Extended Data Fig. 6). RNA, unlike DNA, is highly prone to degradation and is much less likely to be an environmental contaminant. Moreover, this assay did not require to be nested because of the greater amount of template (that is, rRNA versus DNA) per organism. Despite the fact that the majority of samples had not been collected using a protocol optimized for RNA preservation, the level of agreement in classification of GBS by the presence of 16S RNA and DNA was 87%, and this was highly statistically significant ($\kappa = 0.72$, $P = 3.5 \times 10^{-10}$). Moreover, when

we re-analysed the case control study and re-defined GBS status based on the presence of both GBS 16S DNA and rRNA, all of the previously identified associations remained highly statistically significant and the associations with sepsis- and inflammation-related outcomes became stronger (Supplementary Table 8).

### Placental GBS and foetal cytokine storm

Umbilical cord serum was available from 11 samples from term births across both studies where the placenta was positive for GBS DNA. We analysed levels of four pro-inflammatory cytokines that have previously been associated with GBS[8], namely interleukin (IL)-1β, IL-6, IL-8 and tumour necrosis factor (TNF)-α and compared the 11 GBS-positive cases with 129 cord serum samples from infants with GBS-negative placentas where the infants were not admitted to the NNU. Umbilical cord blood levels of IL-1β, IL-6 and IL-8, but not TNF-α, were more likely to be elevated in the group where GBS had been detected in the placenta (Fig. 2b). We defined a foetal cytokine storm as two or more pro-inflammatory cytokines that were elevated >10 times the median level in controls. Cytokine storm was observed in 4/11 (36%) cases where the placenta was GBS positive and 5/129 (4%) placentas were negative for GBS DNA (OR 14.2, 95% CI 3.6–69.8, $P = 0.002$) (Fig. 2c). There was no association between the presence of GBS DNA in the placenta and levels of pro-inflammatory cytokines in placental protein lysates obtained from the same biopsies (Fig. 2d) or in maternal blood before the onset of labour and near term (Fig. 2e).

### Clinical features of cases of proven and probable GBS sepsis

There was a total of 3,689 term births where placental sampling had been performed. Among this group there were three cases of proven GBS sepsis (8.1 (95% CI 1.7–23.7) per 10,000 births), which were all admitted to the NNU. In addition, there were 13 cases of NNU admission with strong clinical or histopathological evidence of sepsis, where GBS DNA was detected in the placenta but not postnatal culture of the infant (35.2 (95% CI 18.8–60.2) per 10,000 births), and we refer to this as probable but culture-negative sepsis. Hence, approximately one birth in 200 had either proven GBS sepsis or probable but culture-negative sepsis plus GBS DNA in the placenta. This is approximately ten times greater than the currently estimated rate of early neonatal GBS disease following term birth[9]. Clinical characteristics for the 16 cases are tabulated separately by whether there was proven GBS sepsis or probable but culture-negative sepsis plus the presence of GBS DNA in the placenta (Table 2). Key observations were that only 3 of the 16 cases had GBS colonization detected in the antenatal period and only three had antibiotics in labour. The majority of the cases were delivered by caesarean section for foetal distress. Interestingly, none of the three cases of proven GBS sepsis had histopathological evidence of inflammation, whereas this was present in the majority of cases of probable but culture-negative sepsis plus the presence of GBS DNA in the placenta.

## Discussion

We performed discovery and validation studies and show that the presence of GBS DNA in the placenta is associated with an increased risk of neonatal morbidity, as evidenced by NNU admission. When analysed by the cause, we found that the association was explained by increased rates of admission with probable but culture-negative sepsis and with

**Fig. 1 | Healthy umbilical cord, healthy membrane roll, funisitis and chorioamnionitis. a–j**, Representative H&E-stained sections of umbilical cords and foetal membranes were scanned using the Zeiss Axio Scan. Histopathological examination included $n = 314$ umbilical cord and $n = 312$ foetal membrane sections. **a**, Low power magnification of a healthy cord. **b**, Medium power magnification of a healthy umbilical artery. **c**, High power magnification of a healthy umbilical vein. **d**, Low power magnification of funisitis. **e**, Medium power magnification of an umbilical artery in funisitis (black arrows indicate extensive

infiltration of neutrophils). **f**, High power magnification of an umbilical vein in funisitis (black arrows indicate neutrophils infiltrating between layers of smooth muscle and yellow arrow indicates microabscess). **g**, Low power magnification of a healthy membrane roll. **h**, Medium power magnification of a healthy membrane roll. **i**, Low power magnification of chorioamnionitis. **j**, Medium power magnification of chorioamnionitis (black arrows indicate extensive infiltration of neutrophils).

culture-positive proven GBS sepsis. Although the number of patients was limited, the validity of these findings was further supported by the fact that there were similar associations when NNU admission was classified on the basis of blinded histopathological examination of the foetal membranes and umbilical cord. Furthermore, the likely mechanism of the observed morbidity in a proportion of these infants

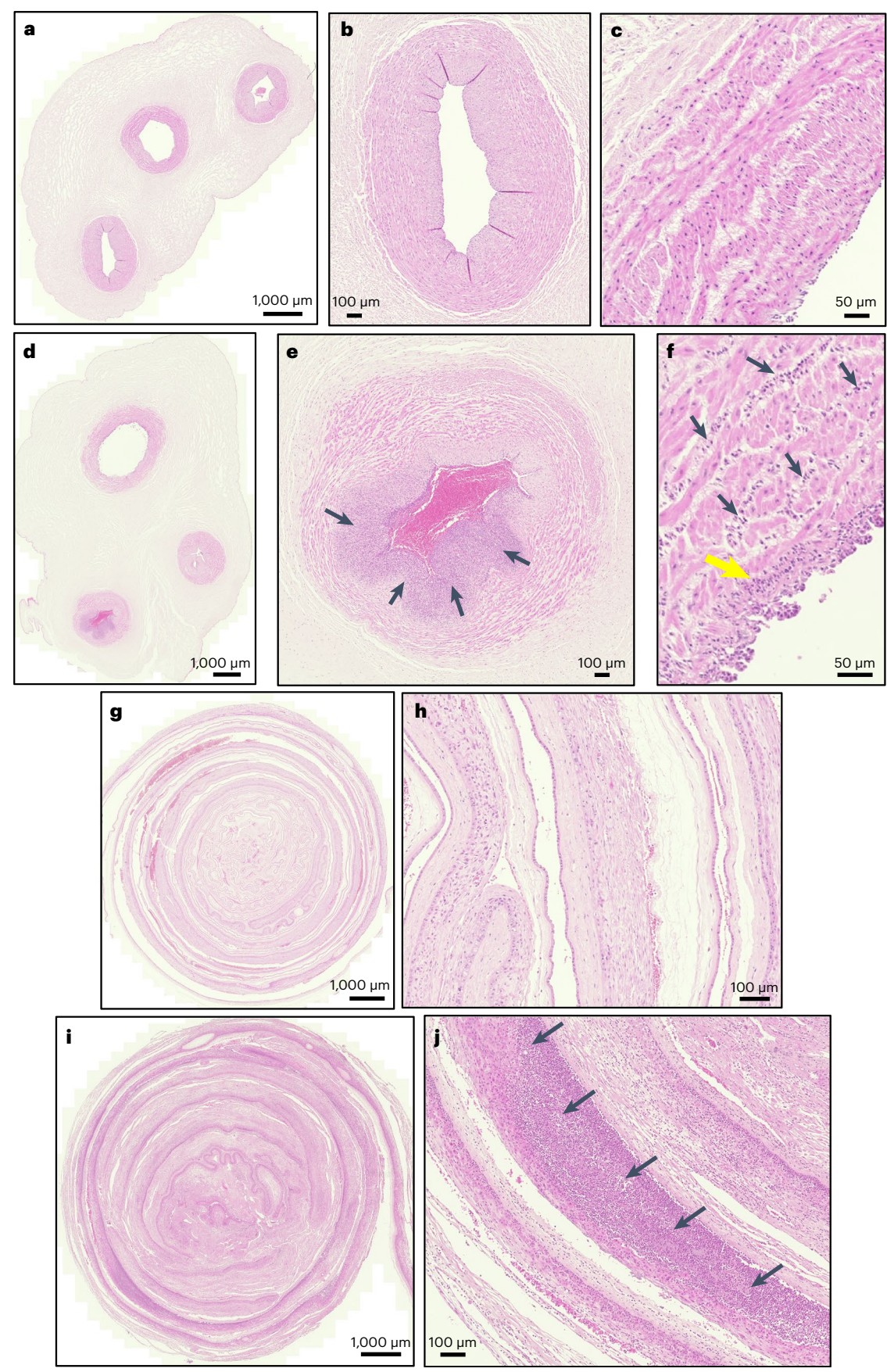

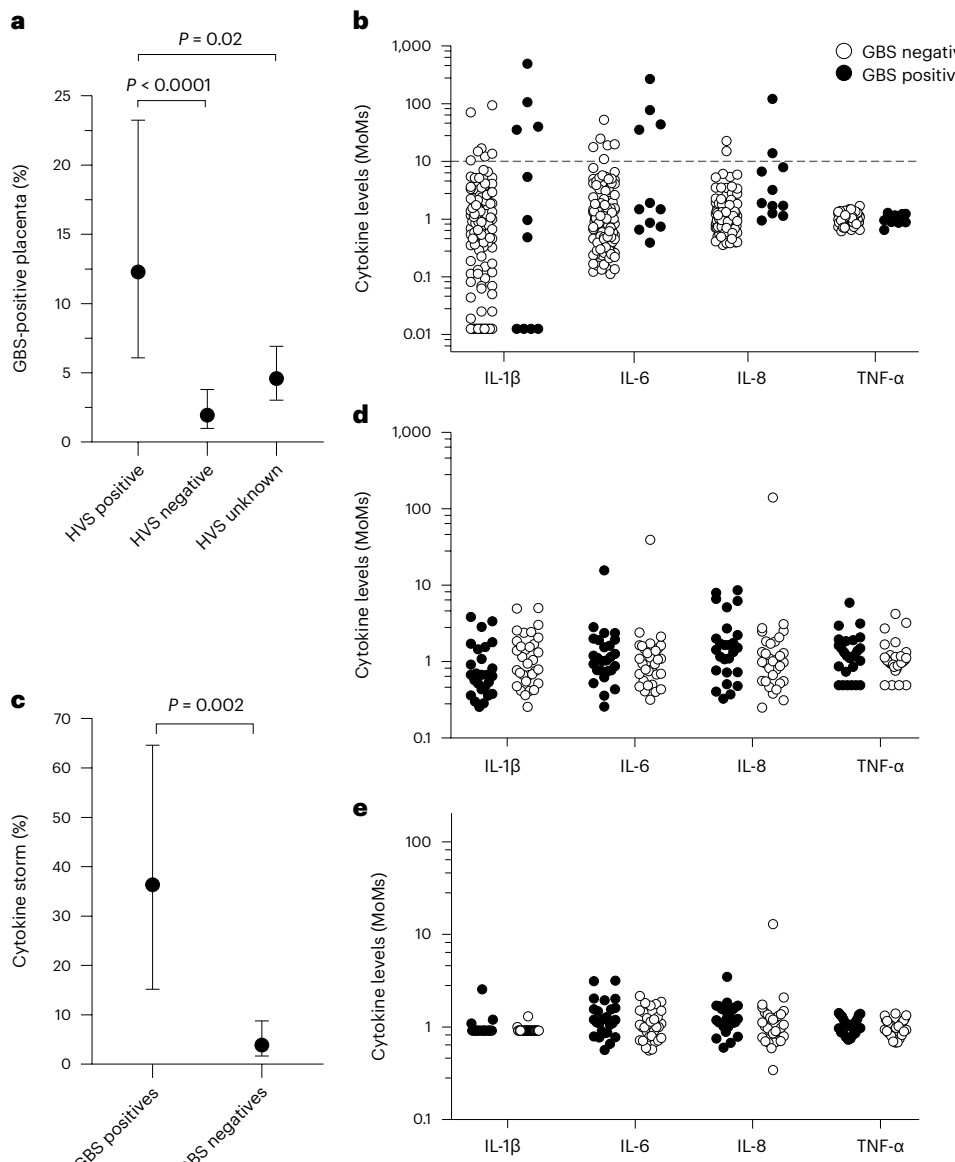

**Fig. 2 | Placental GBS is associated with maternal genital tract colonization and foetal cytokine storm. a**, Proportion of pregnancies with GBS-positive placentas in relation to maternal genital tract colonization assessed by high vaginal swab (HVS), corresponding to 7/57 patients with positive HVS culture, 8/410 patients with negative HVS culture and 21/458 patients with no HVS culture performed. Data are presented as percentage with 95% CI. Two-sided chi-squared test *P* values are reported. **b**, Cytokines were measured in cord serum samples from term pregnancies with GBS-positive (*n* = 11) and negative (*n* = 129) placentas, using the Ella platform (Bio-Techne). Cytokine levels are expressed as the MoM of control samples analysed in the same batch. For graphical purposes, samples with MoM of 0 for IL-1β (*n* = 24) have been plotted using a value of MoM of 0.0126 (the lowest MoM in the dataset). The dotted line represents an elevation of MoM >10. Median cytokine levels in the controls were: 0.18 pg ml⁻¹

for IL-1β (*n* = 116), 16.6 pg ml⁻¹ for IL-6 (*n* = 140), 13.6 pg ml⁻¹ for IL-8 (*n* = 140) and 16.6 pg ml⁻¹ for TNF-α (*n* = 140). **c**, Proportion of pregnancies with foetal cytokine storm in relation to GBS detection in the placenta, corresponding to 4/11 GBS positives and 5/129 GBS negatives. Data are presented as percentage with 95% CI. Two-sided chi-squared test *P* value is reported. **d**, Cytokines were measured as described above in 58 term placental samples from term pregnancies with GBS-positive (*n* = 26) and negative (*n* = 32) placentas. For graphical purposes, samples with MoM of 0 for IL-1β (*n* = 1) and TNF-α (*n* = 9) have been plotted using a value of MoM of 0.256 and MoM of 0.493, respectively (the lowest MoM in the dataset). **e**, Cytokines were measured as described above in 55 maternal serum samples at 36-week gestation from term pregnancies with GBS-positive (*n* = 25) and negative (*n* = 30) placentas. Samples with MoM of 0 for IL-1β (*n* = 49) have been plotted using a value of MoM of 0.914 (the lowest MoM in the dataset).

is a cytokine storm, which was evident in utero. We believe that it is plausible that the presence of a pathogen in a normally sterile site[10] would be causally associated with symptoms, signs and biomarkers of sepsis. The associations were similar when we detected GBS by the presence of GBS rRNA and the presence of GBS RNA suggests the presence of live, albeit rare, bacteria. Moreover, we observed that 10–15% of infants who were born with symptoms, signs or laboratory tests suggestive of sepsis had GBS DNA present in the placenta even though

the organism was not cultured from the neonate. We conclude that GBS causes about ten times the number of cases of neonatal morbidity than is currently recognized and that this morbidity is associated with bacterial invasion of the placenta and extreme activation of the foetal innate immune system before birth.

Among women who are colonized by GBS, the risk of EOD is reduced by more than 80% by the use of intrapartum antibiotic prophylaxis (IAP)[11]. Many countries, including the United States, routinely

**Table 2 | Clinical characteristics of NNU admissions associated with proven GBS sepsis and probable but culture-negative sepsis plus the presence of GBS DNA in the placenta**

| | | Proven GBS sepsis (n=3) | Probable but culture-negative sepsis plus placental GBS DNA (n=13) |
|---|---|---|---|
| **Antenatal** | | | |
| GBS culture (HVS) | Positive, n (%) | 1 | 2 (15) |
| | Negative, n (%) | 0 | 3 (23) |
| | No HVS, n (%) | 2 | 8 (62) |
| Membrane rupture to delivery interval | <24h, n (%) | 3 | 9 (69) |
| | 24–48h, n (%) | 0 | 3 (23) |
| | >48h, n (%) | 0 | 1 (8) |
| **Intrapartum** | | | |
| IAP given | n (%) | 0 | 0 |
| Other IP antibiotics | n (%) | 0 | 3 (23) |
| Indicators of infection | Pyrexia, n (%) | 0 | 5 (38) |
| | Leukocytosis, n (%) | 2 | 10 (77) |
| | Foetal tachycardia, n (%) | 0 | 6 (46) |
| Operative delivery for foetal distress | n (%) | 1 | 6 (46) |
| **Delivery** | | | |
| Gestational age | Median (IQR) | 40.1 (39.0–41.9) | 40.7 (40.1–41.0) |
| Mode | Vaginal, n (%) | 1 | 5 (38) |
| | Caesarean, n (%) | 2 | 8 (62) |
| **Infant** | | | |
| Apgar <7 | 1min, n (%) | 0 | 5 (38) |
| | 5min, n (%) | 0 | 1 (8) |
| | 10min, n (%) | 0 | 1 (8) |
| Acidosis (pH <7.00) | n (%) | 0 | 2 (15) |
| **Histopathology** | | | |
| Chorioamnionitis | n (%) | 0 | 7 (54) |
| Funisitis | n (%) | 0 | 6 (46) |

Proven GBS sepsis was defined as an infant admitted to the neonatal unit with cultured GBS from the septic screen. Probable but culture-negative sepsis with GBS DNA in the placenta was defined as an infant admitted to the neonatal unit where GBS DNA was detected in the placenta by PCR–qPCR and there was clinical evidence of sepsis or histopathology was positive, but no organism was cultured from the septic screen. Note: percentage is omitted for the outcome-proven GBS sepsis, as there were only three cases. Positive GBS culture was a report of a high vaginal swab (HVS) confirming the presence of GBS where the result was reported before the onset of labour. No HVS indicates that the woman did not have a culture result of a vaginal swab in the pregnancy before the onset of labour. Duration of membrane rupture is the interval from the time of membrane rupture to the delivery of the infant, measured in hours. IAP is intrapartum antibiotic prophylaxis, i.e. benzyl-penicillin in non-allergic women. Other intrapartum (IP) antibiotics exclude cases where a single dose of antibiotic was given immediately before caesarean delivery as routine prophylaxis. Pyrexia was defined as any measurement of temperature in labour of >38 °C, leukocytosis was defined as a maternal blood white cell count greater than $15 \times 10^9\,l^{-1}$ within 12 h of delivery, and foetal tachycardia was defined as a foetal heart rate >160 beats per minute for a 10 min duration. Acidosis was defined on the basis of either umbilical venous or arterial blood. Chorioamnionitis and funisitis were defined as described in Methods.

screen all pregnant women for GBS and administer IAP to those who screen positive. A substantial minority of countries (40%), including the United Kingdom, do not perform universal screening and IAP is only given to women who have a positive culture following a clinically

indicated swab[12]. The decision not to screen in the United Kingdom is based on multiple facets of the problem, including the fact that detection of genital tract colonization is temporally variable in a given woman, only a small minority of babies exposed to GBS develop a clinical infection, a very large number of women need to be treated to prevent each case of EOD, there is an absence of level 1 evidence that universal screening improves outcomes, and the rates of EOD in the United Kingdom are comparable with other countries where screening takes place[13]. However, the current analysis demonstrated that only 3/16 cases affected by GBS (that is, proven GBS sepsis or probable but culture-negative sepsis where GBS DNA was present in the placenta) received antibiotics in labour. Given the efficacy of IAP to prevent neonatal sepsis, these results suggest that identification of these cases and treatment with antibiotics in labour may have prevented a number of the adverse outcomes.

The current findings are of clinical relevance in a number of areas. First, the study highlights that analysis of the placenta immediately after birth may allow detection of intra-uterine invasion by GBS and thus infants who are at increased risk of neonatal morbidity could be identified. Second, as GBS was clinically undetectable in the neonate in the majority of cases of neonatal morbidity associated with the presence of GBS DNA in the placenta, and as we have previously shown that GBS DNA is present in the placenta in very small amounts[5], the data indicate that some neonatal morbidity may be due to an exaggerated inflammatory response to a burden of pathogen that is too low to be detected as a clinical infection. Experiments in neonatal mice have demonstrated that administration of small numbers of GBS colony forming units results in an increased risk of neonatal death, which can be prevented by immune blockade (genetic or pharmacological)[14]. The current study is also highly relevant to the development of interventions to prevent GBS sepsis, such as trials of GBS vaccination in pregnancy[15] and a current United Kingdom randomized controlled trial of screening for GBS and IAP (GBS3, ISRCTN49639731; https://www.gbs3trial.ac.uk/). Our findings indicate that the incidence of adverse neonatal outcome that might be prevented by such interventions is much greater than previously thought, being ~1 in 200 as opposed to ~1 per 2,000 births at term. Identifying all cases of neonatal morbidity caused by GBS could make it easier to demonstrate the clinical effectiveness of interventions. Moreover, the potential clinical impact of these interventions may be far greater than estimated on the basis of previous assessments of the burden of disease. The study is also relevant when considering whether physiological development of the foetus includes colonization with microbiota[10]. We found that the only organism present in the placenta before labour onset was GBS[5]. As we now show that the presence of this organism in the placenta is associated with an increased risk of adverse neonatal outcome, the current study provides further support for the 'sterile womb' model of normal foetal development.

The strong association between placental GBS and genital tract colonization is consistent with the possibility that the infection originates from the genital tract. However, only a minority of cases with placental GBS DNA had chorioamnionitis, which suggests that placental infection by GBS is not invariably due to passage of the organism through the foetal membranes. The timing of invasion is probably before the onset of labour in a large proportion of affected placentas, as we previously demonstrated that the proportion of placentas that were positive for GBS DNA did not vary between women who delivered before versus after labour onset and membrane rupture[5]. It is also possible that invasion precedes pregnancy as it is known that GBS can infect the non-pregnant endometrium[16]. Hence it is possible that the GBS detected in the placenta originated from the decidua; that is, placental infection could occur by implantation of the conceptus in GBS-infected decidua, followed by local invasion of the placenta by decidual GBS. Further studies will be required to address this point. It is of particular interest that none of the three cases of proven GBS sepsis

had histopathological features, indicating that infection ascended during labour, whereas two out of the three cases had GBS present in the placenta. Although the numbers are small, these findings indicate that antenatal invasion by GBS may be a particular feature of culture-proven GBS EOD.

Strengths of the current study include the fact that it was prospective, there were hypothesis-generating and hypothesis-testing elements, and we combined extensive clinical data with comprehensive biological samples. The latter allowed objective classification of inflammation in 99% of NNU admissions using histopathology. Moreover, collection of cord blood allowed us to address the mechanisms underlying the associations. However, the study also has weaknesses. First, this is a report from a single centre with a relatively homogeneous population. Second, we did not have genital tract culture results from all participants as universal screening for GBS is not currently recommended in the United Kingdom[17]. However, as discussed above, the data do not support contamination during vaginal delivery as a plausible explanation of our findings. A third relative weakness is that, although we sampled four sites of the placenta, the volume of tissue used was a small fraction of the total organ and, therefore, we may have underestimated the proportion of cases with intra-uterine invasion. However, there was a high level of agreement for the detection of GBS analysing different samples from the same placentas using two different methods. Fourth, we confined the current analysis to term births only. This is because neonatal unit admission and suspected sepsis are very common among infants born preterm; hence, unlike term births, there is not a valid control group of healthy preterm infants. Finally, the immediate clinical application of these results may be limited as it is currently not feasible to test the placenta for the presence of extremely low levels of GBS DNA within a timescale that can inform immediate neonatal care. However, a battery of novel molecular methods for rapid detection of pathogens have been developed, many of which have been evaluated and refined in the coronavirus disease 2019 pandemic[18]. It is plausible that technological developments in this field could be applied in the perinatal setting for the detection of placental GBS.

## Methods

### Patient selection

We used data and samples from a prospective cohort of unselected nulliparous women with a singleton pregnancy and attending the Rosie Hospital, Cambridge UK for antenatal care between 2008 and 2013. The only clinical exclusion criterion for the study was multiple pregnancy. The rationale, design and conduct of the study have been described elsewhere[6,19,20]. In brief, women were recruited around the time of their dating ultrasound scan (typically at ~12 weeks) and were seen for research visits at 20, 28 and 36 weeks of gestational age. Following delivery, a team of technicians performed systematic sampling of the placenta, umbilical cord and foetal membranes, and samples were flash frozen for molecular analysis and fixed for microscopy (see below). Umbilical cord blood was collected from about a third of participants. Outcome data were obtained for all participants by individual examination of the medical record of the mother and by linkage to a range of electronic databases of clinical information, including the results of all microbial cultures performed during the pregnancy. In the whole pregnancy outcome prediction (POP) study population ($n = 4{,}212$), the median age, height and body mass index (BMI) and interquartile ranges (IQRs) were 30.3 (26.8–33.4) years, 165 (161–169) cm, and 24.1 (21.8–27.3) kg m$^{-2}$, respectively, and 13% of the women were smokers at recruitment. In the current work, preterm births were excluded and detailed characteristics of women included in this study are given in Supplementary Tables 1 and 2. Participants received no compensation for taking part in the study. Ethical approval for the study was given by the Cambridgeshire 2 Research Ethics Committee (reference number 07/H0308/163), and all women gave written informed consent. This study is in compliance with all relevant ethical regulations.

### Discovery case control study

Participants eligible for the first case control study were those included in a previous paper[5]. In brief, these participants included two subgroups: (1) participants delivered by planned caesarean section where GBS was detected in the placenta using deep sequencing, and (2) participants where the presence of GBS DNA was determined by 16S rRNA amplicon sequencing, using DNA obtained from the same placenta using two different DNA extraction kits. The participants in subgroup 1 were all delivered by planned caesarean section and consisted of 20 women with a diagnosis of preeclampsia, 20 women delivering a small infant for gestational age and 40 controls. The participants in the second subgroup consisted of 100 women with a diagnosis of preeclampsia, 100 women delivering a small infant for gestational age, 100 preterm births and 200 controls. The women in the second subgroup were not selected in relation to mode of delivery. As there was overlap between the two groups, the two subgroups combined had data on placental GBS status of 537 participants. We excluded preterm births and one sample failed analysis, resulting in a study group of 436 births at term. Case status was defined as admission of the infant to the neonatal unit (NNU) within 48 h of birth and for 48 h or more, which had previously been defined and employed as a marker of significant and immediate neonatal morbidity[6]. Admissions were confined to episodes following delivery (that is, the definition did not include infants who were discharged from hospital following delivery and subsequently readmitted). Controls were pregnancies where this did not occur. The characteristics of the cases and controls are tabulated (Supplementary Table 1). Briefly, the median maternal age varied between 31 and 30 years between cases ($n = 41$) and controls ($n = 395$), respectively. The median BMI was similar between the groups (24 kg m$^{-2}$ in cases and 25 kg m$^{-2}$ in controls). The prevalence of smoking at booking was 15% in cases and 6% among the controls and the prevalence of alcohol consumption was 0% in cases and 5% among the controls. Patient selection for the current work is presented in a flow diagram (Extended Data Fig. 1).

### Validation case control study

Participants eligible for the second case control study were pregnancies where delivery occurred at term, the placenta had been biopsied after the birth and the baby was live born. Cases were defined as any eligible participant where the infant was admitted to the neonatal unit without limit of timing or duration. The scope of neonatal admission was broadened as further analyses to define sepsis phenotypes was planned (see below) but, again, cases related solely to the delivery admission. Controls were selected in a ratio of two controls for each case. After selection of cases and controls, we excluded any participants included in the first case control study. The characteristics of the cases and controls are tabulated (Supplementary Table 2). Briefly, the median maternal age varied between 31 and 30 years between cases ($n = 239$) and controls ($n = 686$). The median BMI was similar between the groups (25 kg m$^{-2}$ in cases and 24 kg m$^{-2}$ in controls). The prevalence of smoking at booking was 5% in both groups and the prevalence of alcohol consumption was 5% in cases and 4% among the controls. Patient selection is presented in a flow diagram (Extended Data Fig. 1).

The cause of neonatal unit admission was defined on the basis of clinical assessment of the infant's medical record and by histopathology of the foetal membranes and umbilical cord. Both forms of assessment were performed blind to placental GBS DNA status. Clinical assessment was performed by an experienced consultant neonatologist (H.S.W.) and admissions were categorized as being unrelated to sepsis (no clinical signs or laboratory tests suggestive of sepsis), possible sepsis (clinical signs of sepsis but no laboratory tests supportive of sepsis), probable sepsis (clinical signs of sepsis plus laboratory tests supportive of sepsis but with a negative septic screen) and proven sepsis (GBS cultured from a normally sterile site) (Supplementary Table 3). Histopathological examination was performed by an experienced

consultant perinatal pathologist (F.J.). Membranes ($n$ = 312) were prepared as a roll and cut in cross section, and umbilical cord ($n$ = 314) was also cut in a cross section where all cord vessels were included. Sections were stained using haematoxylin and eosin (H&E) and reported blind to placental GBS DNA status, on the basis of a previously described classification system[7], where stage 2 or stage 3 evidence was regarded as positive for chorioamnionitis or funisitis, and stage 1 or no evidence was regarded as negative.

### Placental biopsy

Placentas were collected after delivery and the procedure has previously been described in detail[19]. Placental sampling was confined to the placental terminal villi (foetal tissue). Villous tissue was obtained from four separate lobules of the placenta after trimming to remove adhering decidua from the basal plate, that is samples were from the interior of the placenta. Areas of the placenta to be biopsied were selected on the basis of no visible damage, haematomas or infarctions. To remove blood and surface contamination the samples were rinsed in chilled sterile phosphate-buffered saline (Oxoid Phosphate Buffered Saline Tablets, Dulbecco A; Thermo Fisher Scientific dissolved in ultrapure water (ELGA Purelab Classic 18 MΩ cm)). After collection, all placental samples were frozen in liquid nitrogen and stored at −80 °C until further processing. For DNA isolation, approximately 25 mg of villous tissue (combined weight obtained from fragments from all four biopsy collection points) was cut from the stored tissue. Tissue processing was carried out in a class 2 biological safety cabinet.

### DNA analysis in the discovery case control study

DNA isolation and the methods for detecting the presence of GBS DNA in the first case control study were described in detail in the original publication[5]. In brief, DNA was extracted using an MP Biomedical kit. Deep sequencing was performed using the Illumina XTen platform, with 150 base pair paired-end sequencing with an average of 424 million reads per sample. A sample was defined as positive for GBS if there was ≥1 read aligned to the GBS genome. The 16S rRNA PCR was performed using degenerate primers and amplified the V1–V2 region of the gene. Amplicons were sequenced on the Illumina MiSeq platform using 250 base pair paired-end sequencing. Samples were defined as GBS DNA positive if >1% of amplicons were derived from GBS. Analyses were replicated using the threshold >0.1%. The previous study reported a high level of agreement between deep sequencing and 16S rRNA amplicon sequencing ($P = 1.5 \times 10^{-8}$). There were no cases where a sample was positive by 16S rRNA and negative by deep sequencing but there were 4 samples out of 79 that were positive by deep sequencing but negative by 16S rRNA amplicon sequencing. This could either indicate greater sensitivity or more false positives using deep sequencing.

### DNA analysis in validation case control study

DNA was isolated from placental tissue using the QIAamp 96 DNA QIAcube HT Kit (Qiagen; 51331) and the QIAcube HT instrument for automated high-throughput nucleic acid purification in 96-well format (Qiagen). The DNA isolation was carried out according to the manufacturer's instructions with the following initial steps. Placental biopsies were lysed in 200 µl of lysis buffer, containing 20 mM Tris–HCl (pH 8) with 2 mM ethylenediaminetetraacetic acid (EDTA) (Sigma; T9285), 1.2% Triton X-100 (Sigma; T8787), lysozyme 10 mg ml⁻¹ (Sigma; 10837059001), mutanolysin 300 U ml⁻¹ (Sigma; SAE0092), and incubated at 37 °C for 30 min with 400 rpm shaking. After addition of 20 µl of proteinase K per sample (provided by the QIAamp 96 DNA QIAcube HT Kit), samples were incubated at 56 °C overnight with 600 rpm shaking. The following day the samples were transferred to the 96-well plate and 4 µl of RNase A (100 mg ml⁻¹) was added to each sample (Qiagen; 19101). After an incubation at room temperature for 5 min, the plate was loaded on the QIAcube HT instrument and run using the protocol for DNA isolation with the final elution in 200 µl of

elution buffer performed twice. Negative controls performed at this stage had tissue lysis buffer carried through the entire experimental workflow ($n$ = 17 DNA extraction blanks; Extended Data Fig. 5). DNA concentrations were determined by Nanodrop Lite (Thermo Fisher Scientific) and samples were diluted to a concentration of 12.5 ng µl⁻¹ with IDTE (10 mM Tris and 0.1 mM EDTA, pH 8) (Integrated DNA Technologies; 11-05-01-05), using the PIPETMAX Liquid Handler (Gilson).

The presence of GBS DNA in placental samples was assessed using a nested multiplex PCR–qPCR assay targeting the GBS *sip* and *16S* rRNA genes, and the human RNaseP gene (*RPPH1*). Briefly, the first round (outer) of PCR was performed using the Multiplex PCR 5X Master Mix (New England Biolabs; M0284S). The *sip* primers and probe were as previously described[5]. The PCR–qPCR assay for the *16S* rRNA gene utilized a GBS-specific region of the gene (Extended Data Fig. 2). The following primers were employed for detection of the GBS *16S* rRNA gene: forward (outer F) 5′-TAAAAGGAGCAATTGCTTCACTGTG-3′ and reverse (outer R) 5′-TCCTCCAGTTTCCAAAGCGT-3′; amplification was performed in 50 µl with primers at a final concentration of 0.2 µM and 500 ng of placental DNA per reaction. Negative and positive controls performed at this stage had H₂O ($n$ = 25 reactions) or GBS genomic DNA ($n$ = 100 reactions with different GBS gDNA quantities, that is, 4,000, 400, 40 or 4 copies per well; ATCC; BAA-611DQ) as the reaction substrates, respectively (Extended Data Fig. 5). The PCR amplification profile had an initial step of 95 °C for 3 min followed by 15 cycles of 95 °C (20 s), 48 °C (60 s) and 68 °C (60 s) and a final incubation at 68 °C for 3 min. The second round (nested) of qPCR was performed using the TaqMan Multiplex Master Mix (Thermo Fisher Scientific; 4461882); the sip TaqMan assay as previously described with FAM dye/MGB probe at a final 1× concentration (Thermo Fisher Scientific, Assay ID: Ba04646276_s1); a custom-made TaqMan assay for the GBS *16S* rRNA gene at a final 1× concentration with forward primer (inner F) 5′-AATGGACGGAAGTCTGAC-3′, reverse primer (inner R) 5′-GTTAGTTACCGTCACTTGGTA-3′ and JUN dye/QSY probe 5′-AGAGAAGAACGTTGGTAGGAGTGGA-3′ (primers and probe by Thermo Fisher Scientific); and a human RNaseP TaqMan assay with VIC dye/MGB probe (gene symbol: *RPPH1*, Thermo Fisher Scientific, Assay ID: Hs04930436_g1) at a final 0.5× concentration. In each well, 6.1 µl of the first round of PCR (or water in the no-template control/blank wells) was used as the reaction substrate in a total volume of 10 µl. Negative and positive controls performed at this stage had H₂O ($n$ = 28 wells) or GBS gDNA ($n$ = 112 wells with different GBS gDNA quantities, that is, 4,000, 400, 40 or 4 copies per well) as the reaction substrates, respectively (Extended Data Fig. 5). The qPCR amplification profile had an initial step of 95 °C for 20 s followed by 50 cycles of 95 °C (5 s), 50 °C (10 s) and 60 °C (20 s). Samples were excluded from the PCR–qPCR analysis if the RNaseP signal was not detectable, indicating absence of gDNA in the PCR assay or a technical failure. The specificity of this assay was tested by amplifying GBS (*Streptococcus agalactiae*) DNA in presence of genomic DNA from *Streptococcus pyogenes* (ATCC; 700294DQ) and *Streptococcus pneumoniae* (ATCC; 700669DQ) (Supplementary Table 9). qPCR data were collected using the instrument software (QuantStudio 6 Flex system version 1.3, Thermo Fisher Scientific).

When we compared samples that had a detectable signal for GBS, there was a very strong correlation between the signals for 16S and sip when the cycle threshold (Ct) count was <30 and no correlation was observed with higher values (Extended Data Fig. 7). There were eight placental biopsies that were positive using the 16S PCR–qPCR assay but negative with sip, and no samples were positive by sip but negative by 16S. Moreover, where a sample was positive using both methods, the Ct count was higher (that is, a weaker signal) for the sip than for the 16S assay (mean difference 2.4, 95% CI 1.0–3.7, $P = 0.0009$). Extended Data Fig. 8 illustrates a representative example. Given that the GBS genome has seven copies of the *16S* rRNA gene but only one copy of *sip*, we assume that there is more template for the 16S assay for a given number

of colony forming units than for the sip assay. We concluded that the 16S PCR–qPCR assay was more sensitive and defined a positive sample simply on the basis of being positive by 16S PCR–qPCR.

### Analysis of GBS 16S rRNA

Placental RNA was isolated from 72 samples (36 GBS DNA-positive and 36 GBS-negative placentas from the validation study) using the RNeasy Plus Mini Kit (Qiagen; 74134) according to the manufacturer's instructions with the following additional steps. Placental biopsies (approximately 15 mg per sample, including fragments from all four biopsy collection points) were lysed in 600 μl of Buffer RTL Plus and homogenized by bead-beating (Lysing Matrix A tubes, 2 × 30 s, speed of 6.0 m s$^{-1}$ on a FastPrep-24, MP Biomedical). The subsequent steps were performed following the manufacturer's instructions and RNA was eluted from the spin columns with 30 μl of RNase-free H$_2$O. RT reactions were performed using 1 μg RNA per sample, 0.1 μM specific primer (outer reverse primer for GBS 16S rRNA described above) and Superscript IV (Invitrogen; 18090050) in a total volume of 25 μl. Incubations were carried according to the instructions provided in the Superscript IV datasheet (65 °C for 5 min, ice incubation for 2 min, 53 °C for 5 min and 80 °C for 10 min). The RT setup included two types of negative control (omitting the enzyme or the template RNA from the reaction) and three positive controls with decreasing concentrations of GBS RNA (100, 20 and 4 fg per well). The qPCR was performed using 3.6 μl of cDNA, the custom-made GBS 16S TaqMan assay described above and the TaqMan Multiplex Master Mix in a total volume of 12 μl. The qPCR amplification profile had an initial step of 95 °C for 20 s, followed by 50 cycles of 95 °C (5 s) and 60 °C (25 s). Samples were run in duplicate and were considered GBS positive if both wells had 16S signal. The qPCR setup included negative and positive controls with H$_2$O or GBS gDNA (4000, 400, 40 or 4 copies per well, as described above) as the reaction substrates, respectively (Extended Data Fig. 5). Analysis of the controls performed at the various steps of the experimental workflow revealed the expected results: no 16S signal in the 20 wells corresponding to the negative controls and signal in 26/28 positive controls (signal was not detected in 2/4 wells corresponding to the RT positive controls obtained from the lowest GBS RNA quantity, that is, 4 fg per well).

### Assessment of contamination

Assessment of contamination in the discovery case control study is described in detail in the original publication[5]. In the validation study we performed an extensive series of control experiments to determine whether the results might be affected by contamination, given that PCR–qPCR assays are highly sensitive and prone to contaminant dependent signals. First, we compared rates of GBS DNA in the placenta in relation to mode of delivery and found that it was less common among infants delivered vaginally (Supplementary Table 5). This is the opposite pattern to what would be expected if the placenta was usually positive due to contamination from the mother's genital tract. Second, we compared rates of GBS DNA in the placenta in relation to week of delivery. The number of GBS-positive placentas demonstrated no temporal association with the week of sampling (Extended Data Fig. 4 and Supplementary Table 4), whereas we previously demonstrated a temporal association with the contaminant signal *Deinococcus geothermalis*[5]. Third, all qPCR plates included analysis of positive and negative controls performed throughout the experimental workflow and each plate was designed to have similar numbers of cases and controls (Extended Data Fig. 5 and Supplementary Table 7). Analysis of signals by plate demonstrated no pattern in the distribution of positive results. Moreover, the signals from positive and negative controls demonstrated 100% agreement with the expected results. Finally, we re-tested all of the GBS DNA-positive placentas plus the same number of controls using different biopsies and detected GBS by RT–qPCR of 16S ribosomal RNA, and then assessed the level of agreement between the

two assays and the associations with the clinical outcomes of interest (Supplementary Table 8).

### Cytokine ELISAs

Cytokine levels were measured using targeted enzyme-linked immunosorbent assays (ELISAs) in umbilical cord and maternal serum samples, and were measured in placental protein lysates using the Ella platform (Bio-Techne) and cartridges for the analysis of IL-1β, IL-6, IL-8 and TNF-α (Simple Plex Cartridge Kit for 32 samples, containing IL-1β/IL-1F2, IL-6 second generation, IL-8, TNF-α second generation; SPCKC-PS-003229, Bio-Techne). Positive controls (human IL-1β; 894962, human IL-6; 894968, human IL-8/CXCL8; 894950, and human TNF-α; 894977, Bio-Techne) were prepared according to the manufacturer's instructions. We ran 145 unique cord serum samples from term pregnancies in two batches and cytokine levels were expressed as the multiple of the median (MoM) of GBS-negative samples analysed in the same batch. In cases where the median was zero, the lowest detected value was used instead of the median in the calculation. GBS positives were defined as samples with 16S signal using the nested PCR–qPCR assay and/or with one or more GBS read identified by deep sequencing and/or GBS 16S rRNA amplicons >1% in the placenta. Positive samples that were weakly positive (that is, GBS 16S rRNA amplicons >0% and <1%; $n = 5$) were excluded from the primary analysis. However, we performed a sensitivity analysis where the process was repeated defining GBS positivity as GBS 16S amplicons >0.1% of all reads (Extended Data Fig. 9). Controls were pregnancies with GBS-negative placentas on all tests completed and with infants not admitted to NNU. Therefore, the current analysis included 140 samples, 11 cases and 129 controls.

Cytokines were also assayed in maternal plasma and in placental biopsies from patients included in the current analysis using the targeted ELISAs described above. A total of 55 maternal serum samples obtained at 36 weeks gestation were assayed, which were randomly selected among patients with GBS-positive ($n = 25$) and negative ($n = 30$) placentas. Cytokines were also measured in protein lysates obtained from 58 term placental samples, which were randomly selected among patients with GBS-positive ($n = 26$) and negative ($n = 32$) placentas. GBS positives and negatives were defined as in the cord serum analysis described above.

Serum samples were diluted 1:2 with sample diluent (SD13) and run according to the manufacturer's instructions. Placental protein lysates were prepared from the same placental biopsies used for the detection of GBS DNA (5–10 mg from each placenta). Tissues were homogenized in 400 μl of lysis buffer (Phosphate Buffered Saline with 0.1% Triton X-100 and protease inhibitors (cOmplete Mini EDTA-free Protease Inhibitor Tablet; Roche; 04693159001)) by bead-beating in Lysing Matrix D tubes (MP Biomedical; 116913050-CF) for 10 s at a speed of 4 m s$^{-1}$ on a FastPrep-24 (MP Biomedical). Lysates were centrifuged twice at 10,000g for 10 min at 4 °C and the supernatants were quantified using the Pierce BCA Protein Assay Kit (ThermoFisher; 23227). Placental lysates were not diluted with SD13 before loading them on the cartridges.

### Statistics

All the aspects of the POP study, including the results of the research ultrasound scans, were conducted blind, and data were unblinded only at the statistical analysis stage. Participants were allocated into groups based on pregnancy outcome and during the experiments analysis batches contained samples from both cases and controls. Outcome data were ascertained by review of each woman's paper case record by an academic clinical fellow (K.S.) and a neonatologist (H.S.W.), and by record linkage to clinical electronic databases[20]. Proportions were compared using the chi-squared test or Fisher's exact test, as appropriate, and Wilson 95% CIs were calculated[21]. Ninety-five percent CI for unadjusted odds ratios were calculated using the combination of the Baptista–Pike method and the mid-*P* approach[22]. Odds ratios adjusted

for maternal characteristics (age, BMI, smoking and marital status) were estimated using multivariable logistic regression. The sample size for the POP study cohort was pre-determined by the total recruitment achieved. Power calculations were performed for all analyses using the available sample size to estimate the statistical power to detect a given strength of association based on two-sided $\alpha = 0.05$. The power calculation for the primary outcome (admission of the infant to the neonatal unit) was based on observing the same relative risk and proportions of exposure and outcome in the validation study as observed in the development study. Power calculations for all the secondary outcomes (foetal cytokine storm, probable and proven sepsis, chorioamnionitis and funisitis) were based on the observed numbers of GBS-positive and GBS-negative placentas in the validation study, the observed proportion of the given outcome in the GBS-negative group of the validation study, and a range of possible odds ratios. The results were plotted for each of the secondary outcomes (Extended Data Fig. 10). All $P$ values were two sided, and all analyses were performed using Stata version 17 (StataCorp LLC) and GraphPad Prism version 9.2.0 (GraphPad Software LLC).

### Reporting summary

Further information on research design is available in the Nature Portfolio Reporting Summary linked to this article.

## Data availability

Any additional data are available from the corresponding authors on reasonable request and subject to a Data Transfer Agreement, as they include clinical information and in compliance with the ethical permission for the POP study. No participant identifiable information will be disclosed. Requests can be addressed to G.C.S.S. (gcss2@cam.ac.uk) and D.S.C.-J. (dscj1@cam.ac.uk) and will be answered within 1 month. Previously described datasets[5] are: the 16S rRNA gene sequencing datasets, which are publicly available under European Nucleotide Archive (ENA) accession number ERP109246 and the metagenomics datasets, which are available with managed access in the European Genome-phenome Archive (EGA) accession number EGAD00001004198. Source data are provided with this paper.

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

## Acknowledgements

The work was funded by the Medical Research Council (United Kingdom; MR/K021133/1) and supported by the National Institute for Health Research (NIHR) Cambridge Biomedical Research Centre (Women's Health theme). K.S. is funded by the Cambridge Wellcome Trust PhD Programme for Health Professionals. We thank L. Bibby, S. Ranawaka, K. Holmes, J. Gill, R. Millar, L. Sánchez Busó, J. Warner and K. Vickers for technical assistance during the study. The views expressed are those of the authors and not necessarily those of the NHS, the NIHR or the Department of Health and Social Care.

## Author contributions

G.C.S.S. and D.S.C.-J. conceived and designed the experiments. F.G., K.S. and S.L. performed the experiments. D.S.C.-J. and M.C.d.G. designed the 16S rRNA PCR–qPCR assay. U.S. performed all patient selection for case control studies and performed statistical analyses. H.S.W. performed blinded assessment of the indication for neonatal unit admission. F.J. performed blinded histopathological examination of the umbilical cord and foetal membranes. K.L.D. provided methodological advice on DNA extraction. J.P. and S.J.P. provided advice on GBS gDNA sequence and detection. E.C. managed sample collection and processing and the biobank in which all sample were stored. All authors contributed to writing the manuscript and approved the final version.

## Competing interests

G.C.S.S. and D.S.C.-J. receive research funding from Pfizer Bacterial Vaccines for studies on the relationship between placental GBS DNA and maternal circulating levels of GBS antibodies. Cambridge Enterprise (United Kingdom) has filed a patent relating to the primers and probe used in the nested PCR–qPCR assay described in this paper with F.G., M.C.d.G., D.S.C.-J. and G.C.S.S. as the named inventors. The competing interests outside the area of the submitted work are as follows: J.P. reports grants from Pfizer and personal fees from Next Gen Diagnostics LLC; S.J.P. reports personal fees from Specific and personal fees from Next Gen Diagnostics; G.C.S.S. and D.S.C.-J. report grants from GlaxoSmithKline Research and Development Limited, grants and non-financial support from Roche Diagnostics Ltd and non-financial support from Illumina Inc; G.C.S.S. reports personal fees from GlaxoSmithKline Research and Development Limited; K.L.D. is an adviser for Pfizer, Minervax and GSK for GBS vaccines; and K.S., U.S., F.J., H.S.W., S.L. and E.C. declare no competing interests.

## Additional information

**Extended data** is available for this paper at https://doi.org/10.1038/s41564-023-01528-2.

**Correspondence and requests for materials** should be addressed to D. Stephen Charnock-Jones or Gordon C. S. Smith.

[1]Department of Obstetrics and Gynaecology, University of Cambridge, Cambridge, UK. [2]Centre for Trophoblast Research (CTR), Department of Physiology, Development and Neuroscience, University of Cambridge, Cambridge, UK. [3]Department of Histopathology, Addenbrooke's Hospital, Cambridge University Hospitals NHS Foundation Trust, Cambridge, UK. [4]Department of Paediatrics, University of Cambridge, Cambridge, UK. [5]Department of Women's and Children's Health, Uppsala University, Uppsala, Sweden. [6]Wellcome Trust Sanger Institute, Hinxton, UK. [7]Tytgat Institute for Liver and Intestinal Research, Amsterdam University Medical Centers, Amsterdam, the Netherlands. [8]Centre for Neonatal and Paediatric Infectious Diseases Research, St George's University of London, London, UK. [9]Department of Medicine, University of Cambridge, Cambridge, UK. [10]Department of Veterinary Medicine, University of Cambridge, Cambridge, UK. [11]These authors contributed equally: Francesca Gaccioli, Katie Stephens, Ulla Sovio. [12]These authors jointly supervised this work: D. Stephen Charnock-Jones, Gordon C. S. Smith. ✉e-mail: dscj1@cam.ac.uk; gcss2@cam.ac.uk

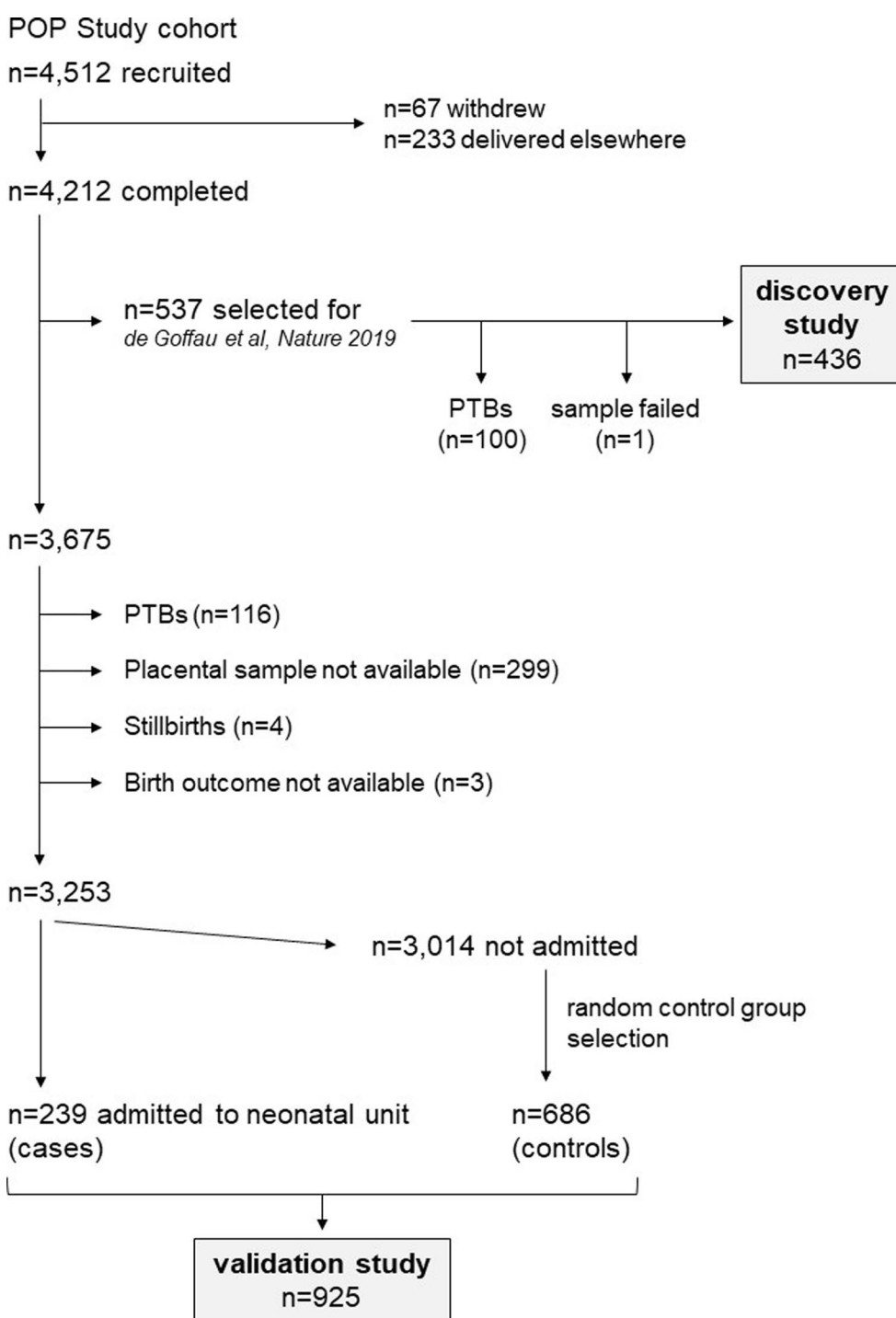

**Extended Data Fig. 1 | Selection of participants for the studies.** Patients in the discovery study have been previously described[5]. Cases (n = 41) and controls (n = 395) are defined as pregnancies with an infant admitted or not admitted, respectively, to neonatal unit within 48 hours of birth and for 48 hours or more. In the validation study cases are defined as pregnancies with an infant admitted to neonatal unit without time limits for admission or duration of stay. The control group is represented by a cohort of random controls selected within the POP study where the baby was not admitted for neonatal care. The validation study excluded any patient who was included in the discovery study. In total, 1,361 patients were included in the current analyses. PTBs, preterm births.

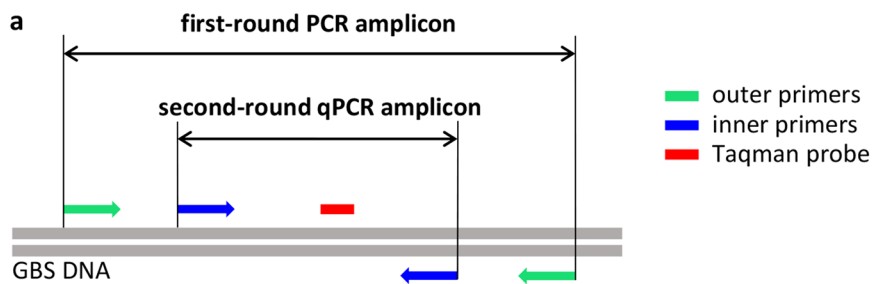

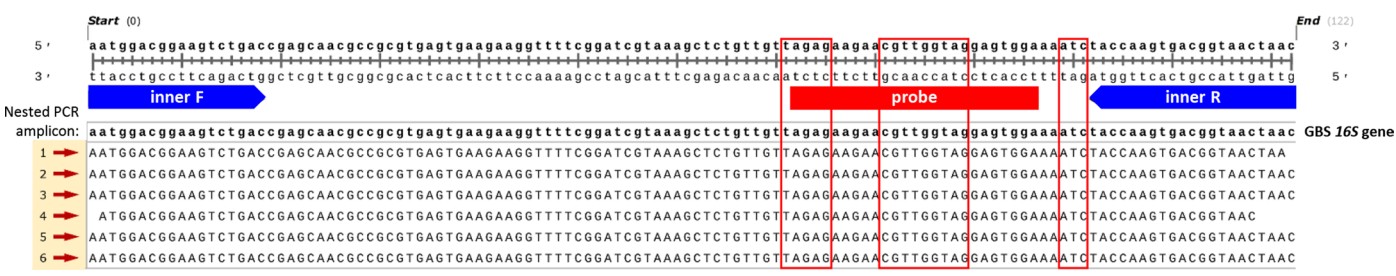

**Extended Data Fig. 2 | Nested PCR-qPCR assay to detect the *16S* GBS gene.**
**a**, Schematic diagram of the nested PCR-qPCR assay. **b**, Multiple sequence alignment of the amplicons generated with the nested PCR-qPCR assay and the *16S* GBS gene sequence (SnapGene Viewer 6.1). The sequences of the PCR amplicons (n = 6 from 4 placentas positive for 16S and sip, and 2 placentas positive for 16S) were determined by Sanger sequencing using the second-round (inner) reverse and forward primers. This analysis confirmed that the sequence of the qPCR products perfectly aligned to the GBS *16S* gene sequence. **c**, Multiple

sequence alignment of the *16S* gene from various *Streptococci*: *S. agalactiae* (GBS) (NR_040821.1), *S. dysgalactiae* (MH393517.1), *S. salivarius* (KM221948.1), *S. anginosus* (MF578782.1), *S. vestibularis* (NR_042777.1), *S. pyogenes* (NR_028598.1), *S. suis* (NR_115737.1), *S. urinalis* (NR_115738.1), *S. equi* (NR_116010.1), *S. uberis* (U41048.1). The position of the inner primers, outer primers and Taqman probe of the nested PCR-qPCR assay are indicated. The red frames highlight regions that differ between species amplified using the qPCR inner primers.

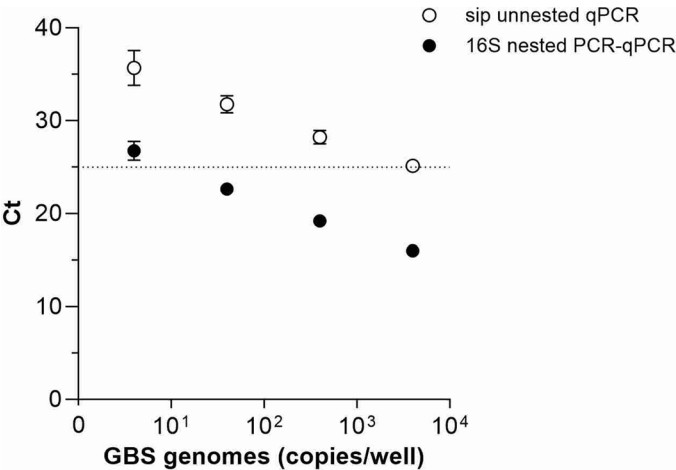

**Extended Data Fig. 3 | Sensitivity of the sip qPCR and 16S PCR-qPCR assays for GBS detection.** Genomic *Streptococcus agalactiae* (GBS) DNA was used at 4000, 400, 40, or 4 copies/reaction in the unnested qPCR and the nested PCR-qPCR assays targeting the *sip* and *16S* GBS genes, respectively. The two curves are described by semi-log equations, which were used to calculate the GBS genome copies at 25 Ct (3,855 and 11 copies required for the sip unnested and 16S nested assay, respectively). Means ± standard deviation are shown; n = 4 independent experiments with 2 technical replicates/each. Ct denotes cycle threshold and it is inversely associated with the relative abundance of the target.

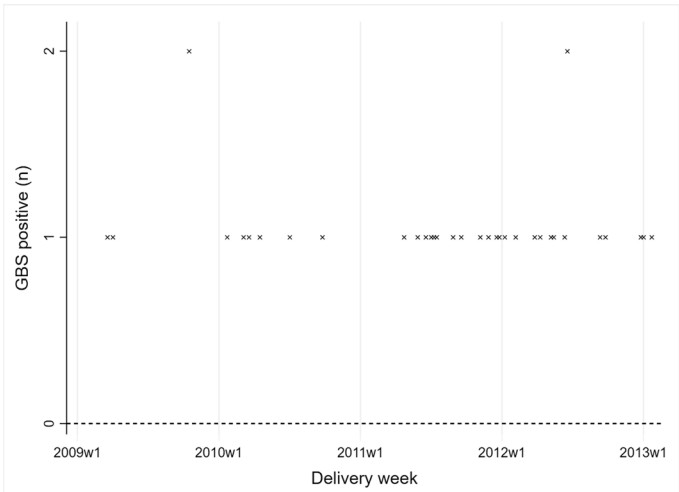

**Extended Data Fig. 4 | Temporal distribution of deliveries with GBS positive placenta in the validation study.** The figure represents the number of weekly deliveries with GBS positive placenta (n = 36 in total). Day 1 of the first week of each year is indicated. n numbers of GBS positive and total deliveries per week are listed in Supplementary Table 4.

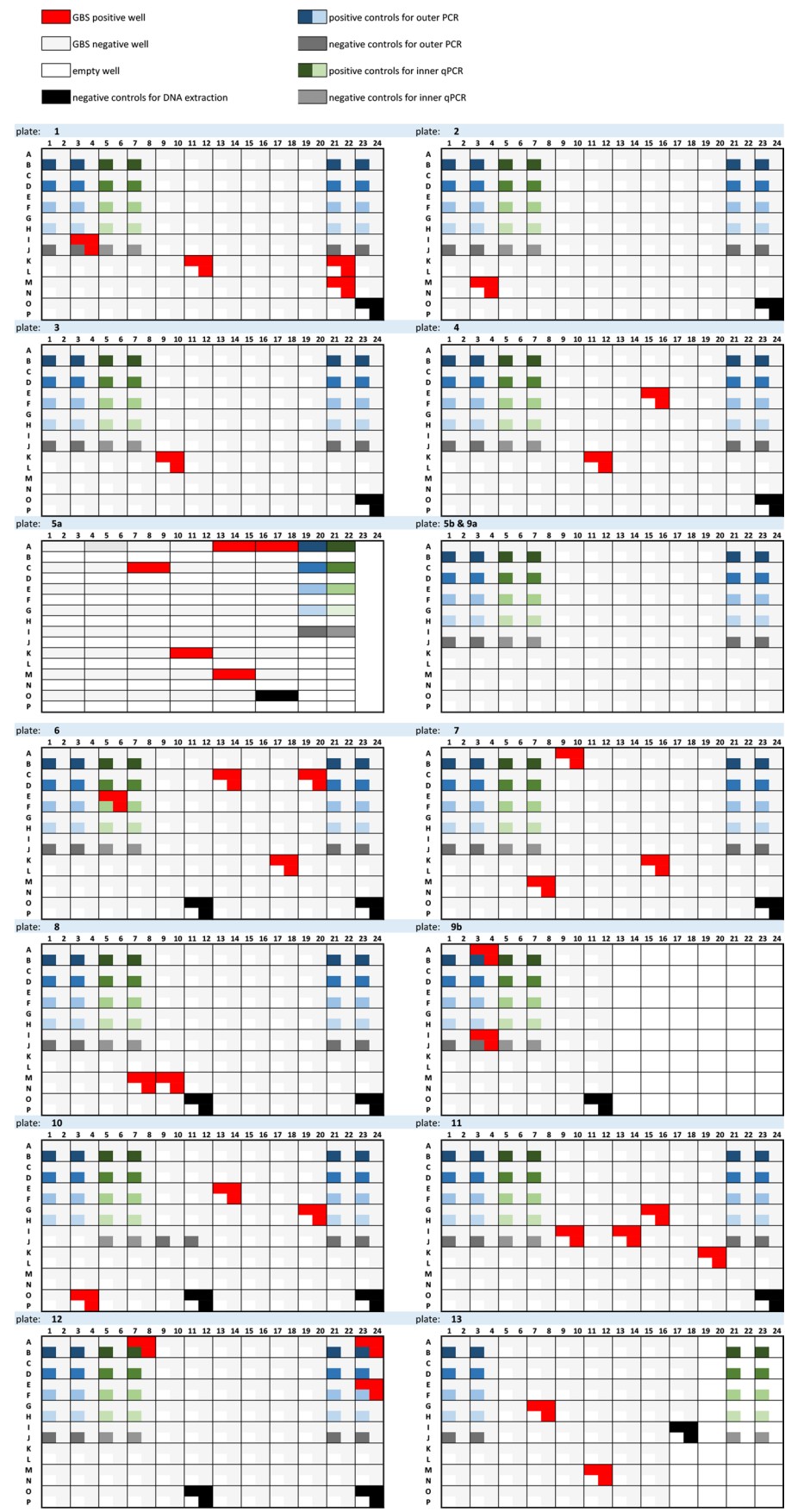

**Extended Data Fig. 5 | See next page for caption.**

**Extended Data Fig. 5 | Spatial distribution of placental samples and controls in the qPCR plates.** The qPCR plates include all the placental samples from the validation study, which were run in triplicate. All the plates had a similar layout with the triplicate wells organized in a L-shape, except for plate 5a which had the triplicate wells organized horizontally. GBS positive samples (in red) had a detectable 16S signal. Positive controls contained decreasing quantities of GBS genomic DNA (4000, 400, 40, or 4 copies/well) added at the outer PCR (blue color gradient) or the inner qPCR (green color gradient) step. Negative controls contained $H_2O$ added at the outer PCR (darker grey) or the inner qPCR (lighter grey) step. DNA extraction blanks (tissue lysis buffer carried through the entire experimental workflow) are indicated in black. Our analysis included: 200 and 112 positive control wells for the PCR and qPCR, respectively, and they all had 16S and sip signal; 50 and 28 negative control wells for the PCR and qPCR, respectively, and they were all 16S- and sip-negative; 51 wells containing DNA extraction blanks, which were all 16S- and sip-negative. In total, we ran 129 negative control wells added at different stages of the experimental workflow and none of them had a detectable signal by either Taqman assay.

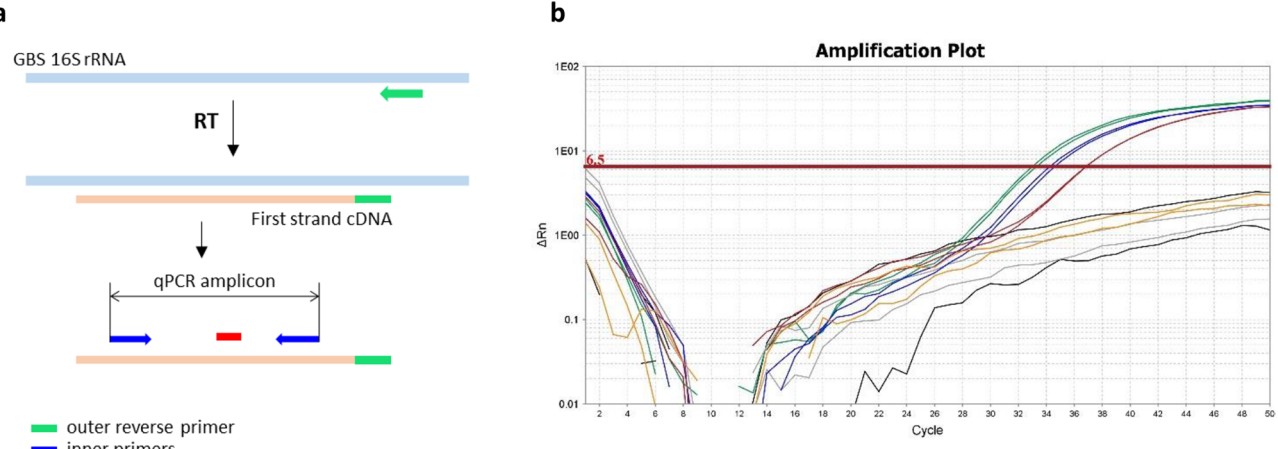

**Extended Data Fig. 6 | RT-qPCR assay to detect GBS 16S rRNA. a**, Schematic diagram of the RT-qPCR strategy. **b**, qPCR amplification plot representing 3 positive samples with signal for GBS 16S rRNA (green, blue and red curves), 1 negative sample with no 16S signal (yellow curve) and 2 negative controls samples (H$_2$O or without the reverse transcriptase enzyme in the RT reaction in black and grey, respectively). Placental samples were run in duplicate and were considered GBS positive if both wells had signal for 16S rRNA. The relative abundance of the target is expressed using the Ct (cycle threshold) value, which is inversely associated with the signal. Rn (normalized reporter value) represents the fluorescence of the reporter dye normalized to the signal of the passive reference dye for a given reaction. The ΔRn is the Rn value of an experimental reaction minus the Rn value of the baseline signal generated by the instrument. This parameter indicates the magnitude of the fluorescent signal generated in the qPCR assay. RT denotes reverse transcriptase.

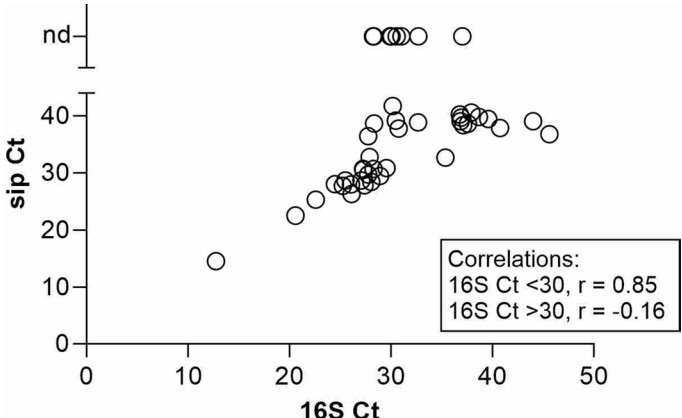

**Extended Data Fig. 7 | Comparison of sip and 16S PCR-qPCR signals for GBS in placental samples.** Correlation between the sip and 16S Ct values obtained in the nested PCR-qPCR assays targeting the two GBS genes (n = 44) using the data from the validation study. Samples with undetectable sip signal are plotted with Ct = nd (not detected, n = 8). Where samples were positive using both methods, the Ct count was higher for sip than for the 16S rRNA PCR-qPCR (mean difference = 2.4, 95% CI = 1.0 to 3.7, P = 0.0009). Pearson's correlation coefficients (r) and two-tailed p values are reported for 16S Ct <30 (p < 0.0001) and Ct >30 (p = 0.56). Ct denotes cycle threshold.

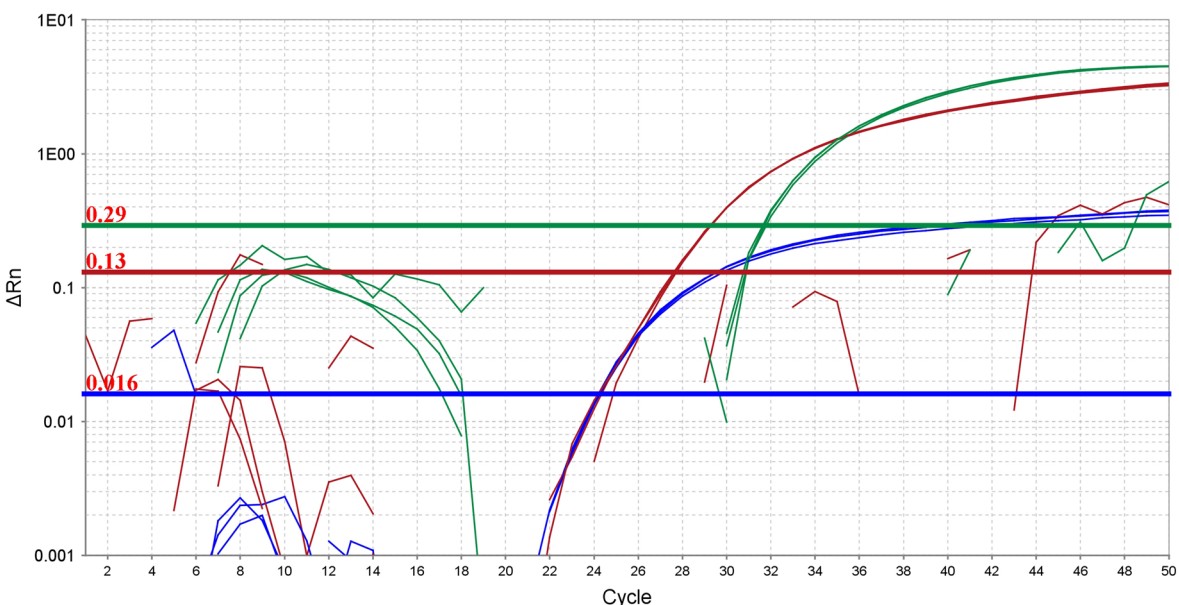

**Extended Data Fig. 8 | Representative amplification plots of the 16S and sip PCR-qPCR assays in a placental DNA sample.** Amplification plots were obtained using the multiplex qPCR assay targeting the *sip* (green) and *16S* (red) GBS genes and the human *RNaseP* (*RPPH1*, blue) gene. The relative abundance of the target is expressed using the Ct (cycle threshold) value, which is inversely associated with the signal. Rn (normalized reporter value) represents the fluorescence of the reporter dye normalized to the signal of the passive reference dye for a given reaction. The ΔRn is the Rn value of an experimental reaction minus the Rn value of the baseline signal generated by the instrument. This parameter indicates the magnitude of the fluorescent signal generated in the qPCR assay.

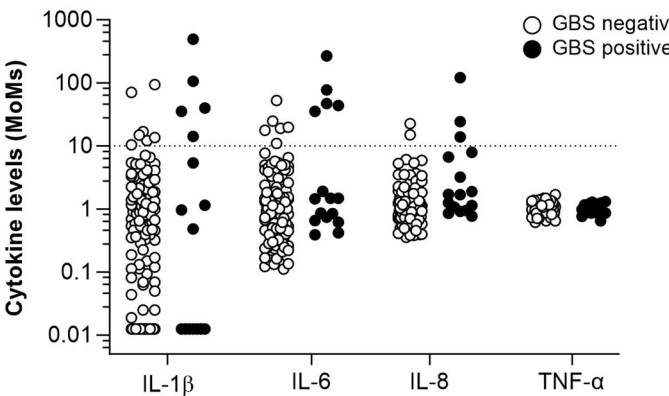

**Extended Data Fig. 9 | Sensitivity analysis of the association between placental GBS and cord cytokine levels.** In this analysis, GBS positives were defined as placental samples with one or more of the following: GBS identified by the 16S PCR-qPCR assay, GBS read identified by deep sequencing, or GBS 16S amplicons >0.1% of all reads (as compared to >1% in primary analysis). Cytokine levels were measured in 145 cord serum samples from term pregnancies with GBS positive (n = 16) and negative (n = 129) placentas, using the Ella platform (Bio-Techne). Cytokine levels are expressed as the multiple of the median (MoM) of control samples analysed in the same batch. For graphical purposes, samples with MoM=0 for interleukin-1β (n = 27) have been plotted using a value of MoM=0.0126 (the lowest MoM in the dataset). GBS negative controls are defined as pregnancies with GBS negative placentas on all tests completed and with babies not requiring NNU admission. The dotted line represents an elevation of MoM >10. IL-1β, interleukin-1β; IL-6, interleukin-6; IL-8, interleukin-8; TNF-α, Tumour Necrosis Factor alpha.

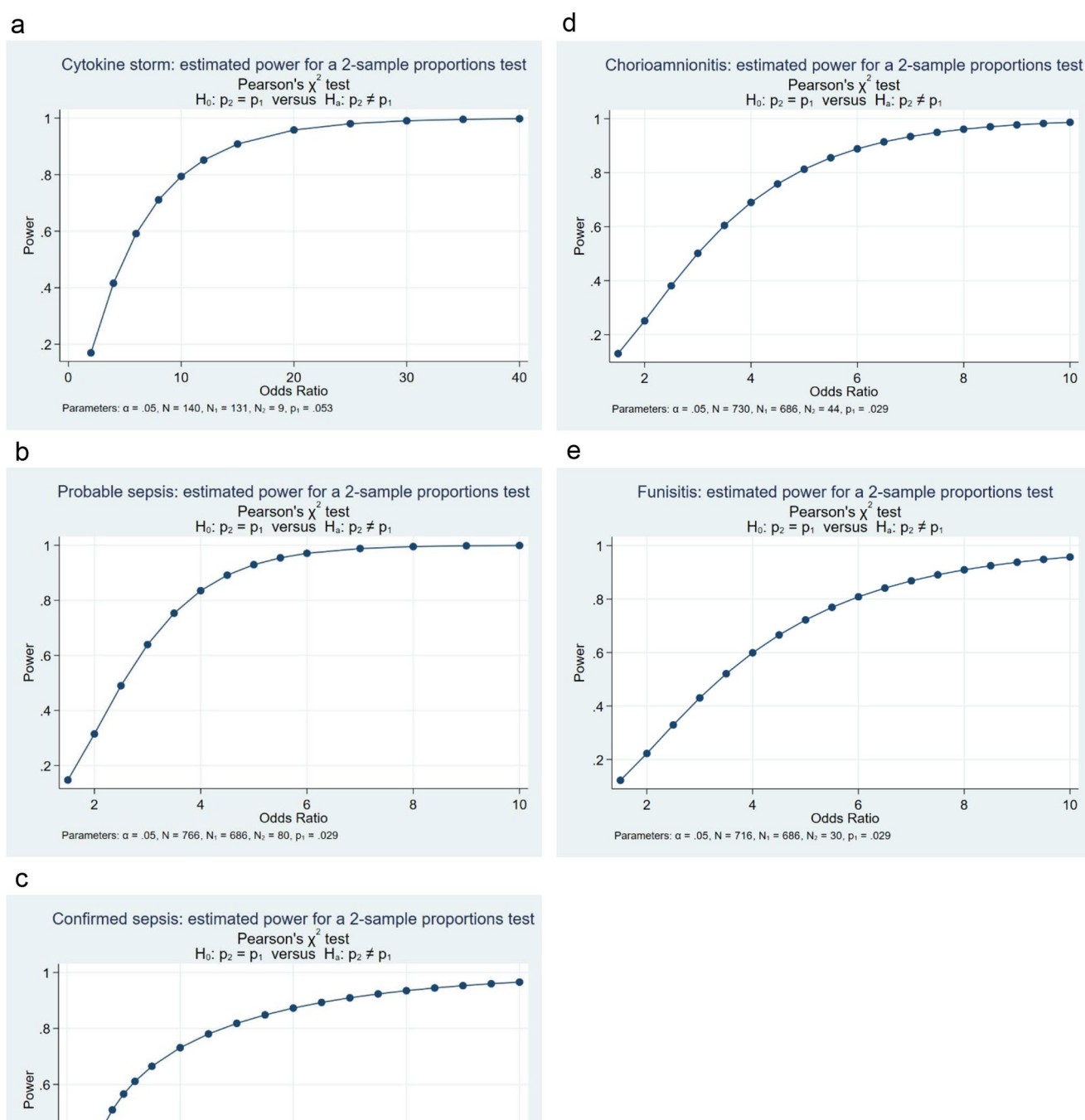

**Extended Data Fig. 10 | Power calculations for the secondary outcomes studied in this work.** Plots illustrate the power calculations for the cytokine storm (**a**), probable (**b**) and proven (**c**) sepsis, chorioamnionitis (**d**) and funisitis (**e**) outcomes and for a range of different odds ratios (OR). Calculations were based on the observed numbers of GBS positive and GBS negative placentas in the validation study, the observed proportion of the given outcome in the GBS negative group of the validation study, and a range of possible odds ratios. Alpha two sided and threshold = 0.05 were applied.

| | |
|---|---|

# Reporting Summary

## Statistics

For all statistical analyses, confirm that the following items are present in the figure legend, table legend, main text, or Methods section.

| n/a | Confirmed | |
|---|---|---|
| ☐ | ☒ | The exact sample size (*n*) for each experimental group/condition, given as a discrete number and unit of measurement |
| ☐ | ☒ | A statement on whether measurements were taken from distinct samples or whether the same sample was measured repeatedly |
| ☐ | ☒ | The statistical test(s) used AND whether they are one- or two-sided *Only common tests should be described solely by name; describe more complex techniques in the Methods section.* |
| ☐ | ☒ | A description of all covariates tested |
| ☐ | ☒ | A description of any assumptions or corrections, such as tests of normality and adjustment for multiple comparisons |
| ☐ | ☒ | A full description of the statistical parameters including central tendency (e.g. means) or other basic estimates (e.g. regression coefficient) AND variation (e.g. standard deviation) or associated estimates of uncertainty (e.g. confidence intervals) |
| ☐ | ☒ | For null hypothesis testing, the test statistic (e.g. *F*, *t*, *r*) with confidence intervals, effect sizes, degrees of freedom and *P* value noted *Give P values as exact values whenever suitable.* |
| ☒ | ☐ | For Bayesian analysis, information on the choice of priors and Markov chain Monte Carlo settings |
| ☒ | ☐ | For hierarchical and complex designs, identification of the appropriate level for tests and full reporting of outcomes |
| ☐ | ☒ | Estimates of effect sizes (e.g. Cohen's *d*, Pearson's *r*), indicating how they were calculated |

*Our web collection on statistics for biologists contains articles on many of the points above.*

## Software and code

Policy information about availability of computer code

| | |
|---|---|
| Data collection | The software used to collect data were the quantitative PCR machine software (QuantStudio 6 Flex system version 1.3, ThermoFisher Scientific) and the Ella platform software version 3.7.2.0 (Bio-Techne). |
| Data analysis | Statistical analyses were performed using Stata version 17 (StataCorp LLC) and GraphPad Prism version 9.2.0 (GraphPad Software LLC). |

For manuscripts utilizing custom algorithms or software that are central to the research but not yet described in published literature, software must be made available to editors and reviewers. We strongly encourage code deposition in a community repository (e.g. GitHub). See the Nature Portfolio guidelines for submitting code & software for further information.

## Data

Policy information about availability of data

All manuscripts must include a data availability statement. This statement should provide the following information, where applicable:
- Accession codes, unique identifiers, or web links for publicly available datasets
- A description of any restrictions on data availability
- For clinical datasets or third party data, please ensure that the statement adheres to our policy

Source data are provided with this paper. Any additional data are available from the corresponding authors at reasonable request and subject to a Data Transfer Agreement, as they include clinical information and in compliance with the ethical permission for the POP study. No participant identifiable information will be disclosed. Requests can be addressed to GCSS (gcss2@cam.ac.uk) and DSC-J (dscj1@cam.ac.uk) and will be answered within 1 month.  Previously described

datasets5 are: the 16S rRNA gene sequencing datasets, which are publicly available under European Nucleotide Archive (ENA) accession number ERP109246; the metagenomics datasets, which are available with managed access in the European Genome-phenome Archive (EGA) accession number EGAD00001004198.

# Human research participants

Policy information about studies involving human research participants and Sex and Gender in Research.

| Reporting on sex and gender | The data reported in this study are related to samples from women of child-bearing age and their infants. The manuscript does not include any analysis based on the biological sex of the babies. |
|---|---|
| Population characteristics | Samples were from the Pregnancy Outcome Prediction (POP) study. In the whole POP study population (n=4212), the median age, height and BMI (IQR) were 30.3 (26.8 to 33.4) years, 165 (161 to 169) cm, 24.1 (21.8 to 27.3) kg/m2, respectively, and 13% of the women were smokers at recruitment. Detailed characteristics of women whose samples were selected for this study are given in Supplementary Tables 1 and 2. Participants received no compensations for being part of the study. In the discovery study (n=436), the median maternal age varied between 31 and 30 years between the groups of 41 cases and 395 controls. The median BMI was similar between the groups (24kg/m2 in cases and 25kg/m2 in controls). The prevalence of smoking at booking was 15% in cases and 6% among the controls and the prevalence of alcohol consumption was 0% in cases and 5% among the controls. In the validation study (n=925), the median maternal age varied between 31 and 30 years between the groups of 239 cases and 686 controls. The median BMI was similar between the groups (25kg/m2 in cases and 24kg/m2 in controls). The prevalence of smoking at booking was 5% in both groups and the prevalence of alcohol consumption was 5% in cases and 4% among the controls. |
| Recruitment | Samples were from the Pregnancy Outcome Prediction (POP) study. Nulliparous women with a viable singleton pregnancy who attended their dating ultrasound scan at the Rosie Hospital (Cambridge, UK) between 14 January 2008 and 31 July 2012 were eligible (n=8028). Of these, 4512 women (56%) provided an informed consent and were recruited. Participants received no compensations for being part of the study. The recruited and non-recruited women were broadly comparable, although according to the hospital record data the women who were recruited were slightly older, more often of white ethnic origin and less likely to smoke. Women were excluded because they delivered elsewhere (n=233) or withdrew their consent (n=67) (Supplementary Figure 1). The cohort of 4212 women used for the sample selection in the present study can be regarded as fairly well representative of the eligible population. See Sovio et al Lancet 2015 PMID 26360240 and Gaccioli et al Placenta 2017 PMCID PMC5701771 for a complete description. |
| Ethics oversight | The Pregnancy Outcome Prediction study was approved by the Cambridgeshire 2 Research Ethics Committee (reference number 07/H0308/163). |

Note that full information on the approval of the study protocol must also be provided in the manuscript.

# Field-specific reporting

Please select the one below that is the best fit for your research. If you are not sure, read the appropriate sections before making your selection.

☒ Life sciences ☐ Behavioural & social sciences ☐ Ecological, evolutionary & environmental sciences

For a reference copy of the document with all sections, see nature.com/documents/nr-reporting-summary-flat.pdf

# Life sciences study design

All studies must disclose on these points even when the disclosure is negative.

| Sample size | A power calculation was performed during the planning phase of the POP study and it is described in Pasupathy et al (BMC Pregnancy and Childbirth 2008 PMID 19019223). In brief, the sensitivity of different models for a given screen positive rate was quantified by 95% confidence intervals. The calculations indicated that the study was likely to provide reasonably precise estimates of sensitivity for conditions with a 3% incidence. The use of a nested case-control design with matching of cases and controls on key maternal characteristics was also planned in advance in the context of expensive or labor intensive methodologies (Pasupathy et al).

The sample size and power calculation for the discovery study have been previously described in detail (de Goffau et al Nature 2019 PMID 31367035).

The validation case-control study (239 cases and 686 controls) included all the eligible participants with a term pregnancy, available placental biopsies and a live born infant admitted to the neonatal unit (without limit of timing or duration). Controls were pregnancies where were the infant was not admitted to the neonatal unit and were selected in a ratio of two controls for each case. Power calculations were performed for all analyses using the available sample size to estimate the statistical power to detect a given strength of association based on alpha (two-sided) = 0.05. The power calculation for the primary outcome (admission of the infant to the neonatal unit) was based on observing the same relative risk and proportions of exposure and outcome in the validation study as observed in the development study. This analysis indicated that we had >99% power to replicate the original finding with the sample size of the validation study. Power calculations for all the secondary outcomes (fetal cytokine storm, probable and confirmed sepsis, chorioamnionitis and funisitis) were based on the observed numbers of GBS positive and GBS negative placentas in the validation study, the observed proportion of the given outcome in the GBS negative group of the validation study, and a range of possible odds ratios (see Supplementary Fig. 10). |
|---|---|

| | |
|---|---|
| Data exclusions | A total of 4512 women with a viable singleton pregnancy were recruited to the POP study. The only clinical exclusion criterion for the study was multiple pregnancy. In the current work preterm births were excluded.<br><br>Samples were excluded from the PCR/qPCR analysis if the RNaseP signal was not detectable, indicating absence of gDNA in the PCR assay or a technical failure.<br><br>In the primary analysis of inflammatory cytokines (Ella, Bio-Techne), positive samples which were weakly positive by 16S rRNA PCR-qPCR (i.e. GBS 16S rRNA amplicons >0% and <1%; n=5) were excluded. |
| Replication | The current work includes 2 non-overlpping cohorts: 1) a discovery case-control study where the initial observation was made, and 2) a validation case-control study where the findings were replicated. |
| Randomization | The POP study is a prospective cohort study of nulliparous women attending the Rosie Hospital (Cambridge, UK) for their dating ultrasound scan. All eligible participants were included. The current work excluded preterm births.<br><br>For the analyses and experiments performed in this manuscript, participants were allocated into groups based on pregnancy outcome (details in Methods). Outcome data were ascertained by review of each woman's paper case record by an Academic Clinical Fellow and a Neonatologist, and by record linkage to clinical electronic databases.<br>During the experiments, batches contain samples from both cases and controls. |
| Blinding | All the aspects of the POP study were conducted blind: the results of the research ultrasound scans and the biochemical marker data were not revealed to the clinicians, patients and researchers performing the downstream experiments.  Classification of cases of NNU admission by evidence for sepsis and histopathological examination of the fetal membranes and the umbilical cord was all performed blind to placental GBS status. In order to evaluate the associations between the exposure (i.e. presence of GBS DNA in the placenta) and the studied clinical outcomes (infant NICU admission, fetal cytokine storm, probable and proven sepsis, chorioamnionitis and funisitis), data were unblinded at the statistical analysis stage. |

# Reporting for specific materials, systems and methods

We require information from authors about some types of materials, experimental systems and methods used in many studies. Here, indicate whether each material, system or method listed is relevant to your study. If you are not sure if a list item applies to your research, read the appropriate section before selecting a response.

## Materials & experimental systems

| n/a | Involved in the study |
|---|---|
| ☒ | ☐ Antibodies |
| ☒ | ☐ Eukaryotic cell lines |
| ☒ | ☐ Palaeontology and archaeology |
| ☒ | ☐ Animals and other organisms |
| ☒ | ☐ Clinical data |
| ☒ | ☐ Dual use research of concern |

## Methods

| n/a | Involved in the study |
|---|---|
| ☒ | ☐ ChIP-seq |
| ☒ | ☐ Flow cytometry |
| ☒ | ☐ MRI-based neuroimaging |

