## [Peer Review File · Nature Microbiology]

Peer Review Information

Journal: Nature Microbiology

Manuscript Title: Placental *Streptococcus agalactiae* DNA is associated with neonatal unit admission and fetal pro-inflammatory cytokines in term infants

Corresponding author name(s): D. Stephen Charnock-Jones, Gordon C. S. Smith

Editorial Notes:

Transferred manuscripts This manuscript has been previously reviewed at another journal that is not operating a transparent peer review scheme. This document only contains reviewer comments, rebuttal and decision letters for versions considered at Nature Microbiology.

Reviewer Comments & Decisions:

Decision Letter, initial version:	
Subject:	Decision on Nature Microbiology manuscript NMICROBIOL-23061561-T
Message:	16th August 2023

Dear Gordon,

Thank you for your patience while your manuscript "Placental *Streptococcus agalactiae* causes fetal cytokine storm and neonatal sepsis" was under peer-review at Nature Microbiology. It has now been seen by 3 referees, whose expertise and comments you will find at the of this email. You will see from their comments below that while they find your work of interest, some important points are raised. We are very interested in the possibility of publishing your study in Nature Microbiology, but would like to consider your response to these concerns in the form of a revised manuscript before we make a final decision on publication. If we are satisfied with your responses, we will not send the revision back to referees.

In particular, you will see that referees #1 and #3 have some relatively minor comments that will need to be addressed with text edits and some additional statistical analyses. The rest referees' reports are clear and the remaining issues should be straightforward to address.

If you have not done so already please begin to revise your manuscript so that it conforms to our Article format instructions at <http://www.nature.com/nmicrobiol/info/final-submission/>

The usual length limit for a Nature Microbiology Article is six display items (figures or tables) and 3,000 words. We have some flexibility, and can allow a revised manuscript at 3,500 words, but please consider this a firm upper limit. There is a trade-off of ~250 words per display item, so if you need more space, you could move a Figure or Table to Supplementary Information.

Some reduction could be achieved by focusing any introductory material and moving it to the start of your opening 'bold' paragraph, whose function is to outline the background to your work, describe in a sentence your new observations, and explain your main conclusions. The discussion should also be limited. Methods should be described in a separate section following the discussion, we do not place a word limit on Methods.

Nature Microbiology titles should give a sense of the main new findings of a manuscript, and should not contain punctuation. Please keep in mind that we strongly discourage active verbs in titles, and that they should ideally fit within 90 characters each (including spaces).

Please include a data availability statement as a separate section after Methods but before references, under the heading "Data Availability". This section should inform readers about the availability of the data used to support the conclusions of your study. This information includes accession codes to public repositories (data banks for protein, DNA or RNA sequences, microarray, proteomics data etc...),

references to source data published alongside the paper, unique identifiers such as URLs to data repository entries, or data set DOIs, and any other statement about data availability. At a minimum, you should include the following statement: “The data that support the findings of this study are available from the corresponding author upon request”, mentioning any restrictions on availability. If DOIs are provided, we also strongly encourage including these in the Reference list (authors, title, publisher (repository name), identifier, year). For more guidance on how to write this section please see: <http://www.nature.com/authors/policies/data/data-availability-statements-data-citations.pdf>

To improve the accessibility of your paper to readers from other research areas, please pay particular attention to the wording of the paper’s opening bold paragraph, which serves both as an introduction and as a brief, non-technical summary in about 150 words. If, however, you require one or two extra sentences to explain your work clearly, please include them even if the paragraph is over-length as a result. The opening paragraph should not contain references. Because scientists from other sub-disciplines will be interested in your results and their implications, it is important to explain essential but specialised terms concisely. We suggest you show your summary paragraph to colleagues in other fields to uncover any problematic concepts.

If your paper is accepted for publication, we will edit your display items electronically so they conform to our house style and will reproduce clearly in print. If necessary, we will re-size figures to fit single or double column width. If your figures contain several parts, the parts should form a neat rectangle when assembled. Choosing the right electronic format at this stage will speed up the processing of your paper and give the best possible results in print. We would like the figures to be supplied as vector files - EPS, PDF, AI or postscript (PS) file formats (not raster or bitmap files), preferably generated with vector-graphics software (Adobe Illustrator for example). Please try to ensure that all figures are non-flattened and fully editable. All images should be at least 300 dpi resolution (when figures are scaled to approximately the size that they are to be printed at) and in RGB colour format. Please do not submit Jpeg or flattened TIFF files. Please see also 'Guidelines for Electronic Submission of Figures' at the end of this letter for further detail.

Figure legends must provide a brief description of the figure and the symbols used, within 350 words, including definitions of any error bars employed in the figures.

- that unprocessed scans are clearly labelled and match the gels and western blots presented in figures.
- that control panels for gels and western blots are appropriately described as loading on sample processing controls

-- all images in the paper are checked for duplication of panels and for splicing of gel lanes.

Please include a statement before the acknowledgements naming the author to whom correspondence and requests for materials should be addressed.

Finally, we require authors to include a statement of their individual contributions to the paper -- such as experimental work, project planning, data analysis, etc. -- immediately after the acknowledgements. The statement should be short, and refer to authors by their initials. For details please see the Authorship section of our joint Editorial policies at http://www.nature.com/authors/editorial_policies/authorship.html

- * include a point-by-point response to any editorial suggestions and to our referees. Please include your response to the editorial suggestions in your cover letter, and please upload your response to the referees as a separate document.

- * ensure it complies with our format requirements for Letters as set out in our guide to authors at www.nature.com/nmicrobiol/info/gta/

- * state in a cover note the length of the text, methods and legends; the number of references; number and estimated final size of figures and tables

- * resubmit electronically if possible using the link below to access your home page: [REDACTED]

- * This url links to your confidential homepage and associated information about manuscripts you may have submitted or be reviewing for us. If you wish to forward this e-mail to co-authors, please delete this link to your homepage first.

Please ensure that all correspondence is marked with your Nature Microbiology reference number in the subject line.

Nature Microbiology is committed to improving transparency in authorship. As part of our efforts in this direction, we are now requesting that all authors identified as 'corresponding author' on published papers create and link their Open Researcher and Contributor Identifier (ORCID) with their account on the Manuscript Tracking System (MTS), prior to acceptance. This applies to primary research papers only. ORCID helps the scientific community achieve unambiguous attribution of all scholarly contributions. You can create and link your ORCID from the home page of the MTS by clicking on 'Modify my Springer Nature account'. For more information please visit www.springernature.com/orcid.

We hope to receive your revised paper within three weeks. If you cannot send it within this time, please let us know.

Yours sincerely,

Reviewer Expertise:

Referee #1: neonatal and maternal health, microbiome and infection [original referee #3 at Nature]

Referee #2: maternal and neonatal health, microbiome and infection [original referee #1 at Nature]

Referee #3: biostatistics, clinical [new referee]

Reviewers Comments:

Reviewer #1 (Remarks to the Author):

The manuscript is improved with clarification on the primary outcomes, discussion of statistical power for achieving this primary outcome, and validation of GBS detection by PCR. However, in many instances the conclusions still seem exaggerated, and do not fit the data and study design.

1. The authors clarify that neonatal unit admission is the primary outcome in the response to reviewers. However, this important detail is not clear. For example, neither the current title (Placental *Streptococcus agalactiae* causes fetal cytokine storm and neonatal sepsis) nor abstract reflect this

critical aspect of the study design. A suggestion for a more appropriate title that describes the data is “Presence of Placental Streptococcus agalactiae DNA is associated with neonatal unit admission in term infants”. Using the word “causes” in the title is inaccurate and misleading, since as the authors point out in the response to reviewers that retrospective analysis can only define “associations”, and not cause and effect relationships.

2. The conclusions stated in the abstract still seem exaggerated and somewhat misleading without clearly stating in the very limited number true GBS bacteremia cases. I would suggest re-focusing the abstract to neonatal unit admissions since this is what this study was designed and powered to evaluate. Mentioning secondary outcomes in the abstract need to highlight the very limited sample size. For example, strong association with proven GBS sepsis (OR=66.7% CI=7.3-963.7) is misleading and should be reworded to state... “among the 3 cases of proven GBS neonatal sepsis, two had placental GBS detection”. Probable sepsis should state, “among the 80 cases of probable neonatal sepsis 10 had placental GBS detection”.

3. Supplementary Table 3. What were the specific parameters in the full blood count, CRP, and chest and abdominal X-ray findings that distinguish between probable and possible sepsis?

4. The different terminologies for infection certainty are not congruent, and make the paper difficult to understand. For example “proven GBS sepsis (abstract)” vs. “confirmed sepsis (line 120)”, vs. “overt GBS (Table 2)”. Since sepsis is a clinical definition, “confirmed GBS bacteremia” should be used instead? More confusing is “probable but culture negative sepsis (abstract)” and “covert GBS (table 2)”. I understand that changing the terminology to overt-covert is limited to only those cases were GBS DNA was detected in the placenta (as opposed to all neonatal unit admissions in the “probable vs. confirmed” comparison). What is especially confusing with this change in definition, is why this change in classification would not also apply to “overt” with limitation by the same criteria (GBS DNA was detected in the placenta).

5. Criteria for discovery and validation. Why were the same criteria not applied with regards to time for admission and duration of stay? Since the focus is on early onset sepsis, would relaxing the case definition for infant admissions “without time limits” still be evaluating early onset sepsis? What is the age cut-off for when “infants” are still admitted to the neonatal unit? (One week? One month? Three months?)

Reviewer #2 (Remarks to the Author):

The authors have addressed the criticisms raised and have improved the manuscript substantially with the addition of the RT-qPCR data on GBS and with better description of controls.

Reviewer #3 (Remarks to the Author):

This is a very nice study assessing the association between placental *Streptococcus agalactiae* and neonatal sepsis. Just one major comment regarding the statistical analyses and a minor comment on Figure 2:

1. Since the outcomes in these studies are fairly common (e.g., >10% occurrence) please consider using modified Poisson regression (rather than logistic) to get prevalence ratios rather than odds ratios (which are difficult to interpret when outcomes are more common).
2. Figures 2a & 2c: please put error bars on these graphs so reader can see variability.

Author Rebuttal to Initial comments

Reviewer #1 (Remarks to the Author):

The manuscript is improved with clarification on the primary outcomes, discussion of statistical power for achieving this primary outcome, and validation of GBS detection by PCR. However, in many instances the conclusions still seem exaggerated, and do not fit the data and study design.

1. The authors clarify that neonatal unit admission is the primary outcome in the response to reviewers. However, this important detail is not clear. For example, neither the current title (Placental *Streptococcus agalactiae* causes fetal cytokine storm and neonatal sepsis) nor abstract reflect this critical aspect of the study design. A suggestion for a more appropriate title that describes the data is "Presence of Placental *Streptococcus agalactiae* DNA is associated with neonatal unit admission in term infants". Using the word "causes" in the title is inaccurate and misleading, since as the authors point out in the response to reviewers that retrospective analysis can only define "associations", and not cause and effect relationships.

R1. We have edited the title to remove the language of causality.

2. The conclusions stated in the abstract still seem exaggerated and somewhat misleading without clearly stating in the very limited number true GBS bacteremia cases. I would suggest re-focusing the abstract to neonatal unit admissions since this is what this study was designed and powered to evaluate.

Mentioning secondary outcomes in the abstract need to highlight the very limited sample size. For example, strong association with proven GBS sepsis (OR=66.7% CI=7.3-963.7) is misleading and should be reworded to state... “among the 3 cases of proven GBS neonatal sepsis, two had placental GBS detection”. Probable sepsis should state, “among the 80 cases of probable neonatal sepsis 10 had placental GBS detection”.

R2. We have added the absolute numbers of events to the abstract.

3. Supplementary Table 3. What were the specific parameters in the full blood count, CRP, and chest and abdominal X-ray findings that distinguish between probable and possible sepsis?

R3. We have added the laboratory and imaging findings which were used to define probable versus possible sepsis (Supplementary Table 3).

4. The different terminologies for infection certainty are not congruent, and make the paper difficult to understand. For example “proven GBS sepsis (abstract)” vs. “confirmed sepsis (line 120)”, vs. “overt GBS (Table 2)”. Since sepsis is a clinical definition, “confirmed GBS bacteremia” should be used instead? More confusing is “probable but culture negative sepsis (abstract)” and “covert GBS (table 2)”. I understand that changing the terminology to overt-covert is limited to only those cases where GBS DNA was detected in the placenta (as opposed to all neonatal unit admissions in the “probable vs. confirmed” comparison). What is especially confusing with this change in definition, is why this change in classification would not also apply to “overt” with limitation by the same criteria (GBS DNA was detected in the placenta).

R4. We have edited the manuscript to use the following definitions consistently:

Cases with positive culture of GBS in the neonatal period are now consistently referred to as “proven GBS sepsis”.

Cases of neonatal unit admission where there was a clinical diagnosis of sepsis (based on clinical and laboratory/imaging indicators) or histopathological evidence of intra-uterine inflammation (chorioamnionitis or funisitis) but no organism was cultured are now consistently referred to as “probable but culture negative sepsis”.

The terms are a bit more cumbersome but we agree with the reviewer that clarity is the priority.

5. Criteria for discovery and validation. Why were the same criteria not applied with regards to time for admission and duration of stay? Since the focus is on early onset sepsis, would relaxing the case

definition for infant admissions “without time limits” still be evaluating early onset sepsis? What is the age cut-off for when “infants” are still admitted to the neonatal unit? (One week? One month? Three months?)

R5. The discovery study used a pre-defined outcome of neonatal unit admission which we have employed in the first output from the POPS cohort in the Lancet in 2015. We used this outcome in the discovery study as it was already defined in our study dataset. Having observed the association with the presence of GBS DNA in the placenta we made a decision to re-analyse all of the other neonatal unit records of the infants admitted to determine the evidence supporting sepsis and to perform blinded histopathology on the placenta and membranes. As the hypothesis being tested was focused on sepsis and we were ascertaining cases by detailed examination of the medical record and biological samples, we included all neonatal unit admissions. All of these cases were still early onset as this analysis was confined to the delivery admission. i.e. we did not include infants who were discharged home and readmitted. We have now clarified this in the methods.

Reviewer #2 (Remarks to the Author):

The authors have addressed the criticisms raised and have improved the manuscript substantially with the addition of the RT-qPCR data on GBS and with better description of controls.

R6. We are grateful to the reviewer for their positive comments.

Reviewer #3 (Remarks to the Author):

This is a very nice study assessing the association between placental *Streptococcus agalactiae* and neonatal sepsis. Just one major comment regarding the statistical analyses and a minor comment on Figure 2:

1. Since the outcomes in these studies are fairly common (e.g., >10% occurrence) please consider using modified Poisson regression (rather than logistic) to get prevalence ratios rather than odds ratios (which are difficult to interpret when outcomes are more common).

R7. We understand the point being made by the reviewer. In analysis of cohort studies with common outcomes, the rare disease assumption is violated and the odds ratio is no longer a reliable approximation of the relative risk (the odds ratio is systematically further from unity than the relative risk and the magnitude of the difference between the two ratios is positively associated with the

frequency of the outcome). However, this analysis is a nested case control study, i.e. we have selected all cases and a sample of controls from the cohort. The outcomes only appear to be common because we analyse all of the cases and a sample of the controls. Conventionally, associations in case control studies are quantified by odds ratios. Please see Chapter 8 (Case-Control Studies) in *Modern Epidemiology Third Edition 2008* (edited by Kenneth J. Rothman, Sander Greenland, Timothy L. Lash). This describes the use of odds ratios in case control studies and on page 114 it states “the rare disease assumption is not needed in case control studies”.

2. Figures 2a & 2c: please put error bars on these graphs so reader can see variability.

R8. We have added bars to indicate the 95% confidence intervals.

Decision Letter, first revision:

Subject: Your manuscript, NMICROBIOL-23061561A

Message: Our ref: NMICROBIOL-23061561A

21st September 2023

Dear Gordon,

Thank you for your patience as we've prepared the guidelines for final submission of your Nature Microbiology manuscript, "Placental Streptococcus agalactiae DNA is associated with neonatal morbidity and fetal cytokine storm at term" (NMICROBIOL-23061561A). Please carefully follow the step-by-step instructions provided in the attached file, and add a response in each row of the table to indicate the changes that you have made. Ensuring that each point is addressed will help to ensure that your revised manuscript can be swiftly handed over to our production team.

In recognition of the time and expertise our reviewers provide to Nature Microbiology's editorial process, we would like to formally acknowledge their contribution to the external peer review of your manuscript entitled "Placental Streptococcus agalactiae DNA is associated with neonatal morbidity and fetal cytokine storm at term". For those reviewers who give their assent, we will be publishing their names alongside the published article.

Nature Microbiology offers a Transparent Peer Review option for new original research

manuscripts submitted after December 1st, 2019. As part of this initiative, we encourage our authors to support increased transparency into the peer review process by agreeing to have the reviewer comments, author rebuttal letters, and editorial decision letters published as a Supplementary item. When you submit your final files please clearly state in your cover letter whether or not you would like to participate in this initiative. Please note that failure to state your preference will result in delays in accepting your manuscript for publication.

Cover suggestions

COVER ARTWORK: We welcome submissions of artwork for consideration for our cover. For more information, please see our [guide for cover artwork](https://www.nature.com/documents/Nature_covers_author_guide.pdf).

Nature Microbiology has now transitioned to a unified Rights Collection system which will allow our Author Services team to quickly and easily collect the rights and permissions required to publish your work. Approximately 10 days after your paper is formally accepted, you will receive an email in providing you with a link to complete the grant of rights. If your paper is eligible for Open Access, our Author Services team will also be in touch regarding any additional information that may be required to arrange payment for your article.

Please note that *Nature Microbiology* is a Transformative Journal (TJ). Authors may publish their research with us through the traditional subscription access route or make their paper immediately open access through payment of an article-processing charge (APC). Authors will not be required to make a final decision about access to their article until it has been accepted. [Find out more about Transformative Journals](https://www.springernature.com/gp/open-research/transformative-journals)

Authors may need to take specific actions to achieve [compliance with funder and institutional open access mandates](https://www.springernature.com/gp/open-research/funding/policy-compliance-faqs). If your research is supported by a funder that requires immediate open access (e.g. according to [Plan S principles](https://www.springernature.com/gp/open-research/plan-s-compliance)) then you should select the gold OA route, and we will direct you to the compliant route where possible. For authors selecting the subscription publication route, the journal's standard licensing terms will need to be accepted, including [self-archiving policies](https://www.nature.com/nature-portfolio/editorial-policies/self-archiving-and-license-to-publish). Those licensing terms will supersede any other terms that the author or any third party may assert apply to any version of the manuscript.

For information regarding our different publishing models please see our [Transformative Journals](https://www.springernature.com/gp/open-research/transformative-journals) page. If you have any questions about costs, Open Access

requirements, or our legal forms, please contact ASJournals@springernature.com.

Please use the following link for uploading these materials: [REDACTED]

Best regards,

Final Decision Letter:

Subject: Decision on Nature Microbiology manuscript NMICROBIOL-23061561B

Message: 16th October 2023

Dear Gordon,

I am pleased to accept your Article "Placental Streptococcus agalactiae DNA is associated with neonatal unit admission and fetal pro-inflammatory cytokines in term infants" for publication in Nature Microbiology. Thank you for having chosen to submit your work to us and many congratulations.

Acceptance of your manuscript is conditional on all authors' agreement with our publication policies (see <https://www.nature.com/nmicrobiol/editorial-policies>). In particular your manuscript must not be published elsewhere and there must be no announcement of the work to any media outlet until the publication date (the day on which it is uploaded onto our website).

Please note that *Nature Microbiology* is a Transformative Journal (TJ). Authors may publish their research with us through the traditional subscription access route or

make their paper immediately open access through payment of an article-processing charge (APC). Authors will not be required to make a final decision about access to their article until it has been accepted. [Find out more about Transformative Journals](https://www.springernature.com/gp/open-research/transformative-journals)

Authors may need to take specific actions to achieve [compliance](https://www.springernature.com/gp/open-research/funding/policy-compliance-faqs) with funder and institutional open access mandates. If your research is supported by a funder that requires immediate open access (e.g. according to [Plan S principles](https://www.springernature.com/gp/open-research/plan-s-compliance)) then you should select the gold OA route, and we will direct you to the compliant route where possible. For authors selecting the subscription publication route, the journal's standard licensing terms will need to be accepted, including [self-archiving policies](https://www.nature.com/nature-portfolio/editorial-policies/self-archiving-and-license-to-publish). Those licensing terms will supersede any other terms that the author or any third party may assert apply to any version of the manuscript.

With kind regards,